# SUMO-dependent transcriptional repression by Sox2 inhibits the proliferation of neural stem cells

**Elisa Marelli**[ORCID]*, **Jaime Hughes**[ORCID], **Paul J. Scotting**

School of Life Sciences, University of Nottingham, Nottingham, Nottinghamshire, United Kingdom

* eli.marelli@gmail.com

**Data Availability Statement:** All relevant data are within the paper and its Supporting Information files.

**Funding:** EM - The present work was funded through BBSRC-Doctoral Training Partnership

## Abstract

Sox2 is known for its roles in maintaining the stem cell state of embryonic stem cells and neural stem cells. In particular, it has been shown to slow the proliferation of these cell types. It is also known for its effects as an activating transcription factor. Despite this, analysis of published studies shows that it represses as many genes as it activates. Here, we identify a new set of target genes that Sox2 represses in neural stem cells. These genes are associated with centrosomes, centromeres and other aspects of cell cycle control. In addition, we show that SUMOylation of Sox2 is necessary for the repression of these genes and for its repressive effects on cell proliferation. Together, these data suggest that SUMO-dependent repression of this group of target genes is responsible for the role of Sox2 in regulating the proliferation of neural stem cells.

## Introduction

All members of the Sox protein family share a high level of sequence similarity and function as transcription factors [1]. They are characterized by a conserved HMG DNA-binding domain composed of 79 amino acids that contains the conserved sequence motif "RPMNAFMVW". Sox proteins are able to recognize and bind to the consensus sequence "5'-WWCAAW-3'" (where W = A/T) [2]. This binding process causes a widening of the minor groove that leads to the bending of the DNA structure [3–5]. It has been proposed that Sox factors function as architectural transcription factors that recruit other protein through their binding to the DNA; the proteins recruited would be other transcription factors, chromatin re-modelers or other regulatory partners that would act as co-activators (or co-repressors) of target genes.

SoxB proteins, which include Sox1, Sox2, Sox3, Sox14 and Sox21, play a central role in several processes during the embryonic development of vertebrates and insects [4, 6–8]. Sox1, Sox2 and Sox3 belong to the SoxB1 subgroup and exhibit a high level of full-length sequence similarity and some functional redundancy. SoxB1 transcription factors are thought to play a major role in the regulation of the pluripotent state of neural stem cells (NSC) in the central nervous system (CNS) [9]. The expression of SoxB1 genes firstly marks ectodermal cells, which have the potential to become neural cells and later expression becomes restricted to ectodermal cells committed to a neural fate. This has been shown in zebrafish, *Drosophila*, *Amphioxus*, *Xenopus* and chicken embryos [10–18].

Programme (PhD) awarded to Elisa Marelli. The funders had no role in study design, data collection and analysis, decision to publish, or preparation of the manuscript.

**Competing interests:** The authors have declared that no competing interests exist.

SoxB1 factors have been shown to play roles in both determining neural fate and in maintaining the multipotency of NSCs [13, 19]. In chicken embryos, the differentiation of neural progenitors was prevented by over-expression of Sox2 and/or Sox3. Importantly, these cells exhibited features of more undifferentiated cells such as the ability to proliferate and the expression of early neural progenitor markers [20, 21]. On the other hand, over-expression of dominant-negative forms of Sox2 and/or Sox3 caused premature exit from the cell cycle and the induction of neuronal differentiation. A study by Kondo *et al.* (2004) showed that through re-initiation of Sox2 expression it is possible to convert mouse oligodendrocyte progenitor cells (OPCs) back into multipotent neural stem-like cells [22]. The same study showed that inhibition of Sox2 caused premature exit from the cell cycle and differentiation of the OPCs towards neural fate [9, 23]. In mouse, Sox2 is expressed in undifferentiated stem and progenitor cells in the CNS and its expression decreases with the progression of differentiation, although it remains expressed in some populations of differentiated neurons [24]. Cavallaro *et al.* (2008) created Sox2 knockdown mutants in mice and studied *in vitro* differentiation of embryonic and adult NSCs. They showed that neuronal differentiation was affected in mutant cells and that this condition could be rescued by over-expression of Sox2 in the early stages of differentiation. According to these data, the activity of Sox2 is therefore crucial for the differentiation of NSC towards neuronal identity. However, it seems that mice are much less affected by defects in the expression of Sox2 compared to humans [25, 26] highlighting how much Sox2 activity and function are context dependent.

Genome-wide single cell analysis, followed by functional analysis in mouse cortex showed that when stem cells transition into rapidly dividing progenitor cells, the expression of Sox2 is reduced [27]. According to this study, stem cells express high levels of Sox2, which represses proliferative genes such as Cyclin D1 through binding of low-affinity DNA motifs. Conversely, as these cells differentiate Sox2 expression is reduced and so expression of Cyclin D1 is derepressed, promoting proliferation.

SoxB1 factors have been generally considered to exert their functions by activating the expression of target genes. However, more recent evidence suggests that SoxB1 factors can be either transcriptional activators or repressors and that their function is highly dependent on the context of when and where they are expressed [28, 29]. This context-dependency could be correlated with post-transcriptional modifications and changes in the interacting partners of the SoxB1 factors.

As for most members of the Sox family, the activity of Sox2, whether it is as an activator or a repressor, is dependent upon the co-factor recruited. In fact, Sox2 is expressed in different developmental and cellular contexts, where it exhibits different activities. Therefore, it is important to define which are the target genes of Sox2 in each context and how Sox2 regulates these targets in different contexts. According to a model proposed by Remenyi *et al.* (2003), Sox2 can partner with a specific co-factor assuming different conformational arrangements depending on the distance between the two binding sites on the DNA [30]. Therefore, Sox2 may elicit tissue-specific functions by switching its interaction partners. However, it is still not clear how these switches are regulated and whether post-translational modifications could play a role in these mechanisms by altering which cofactors Sox2 interacts with.

We postulate that the switch between the activating and repressing activity of Sox2 could be regulated by post-translational modification. In fact, Sox2 is subject to a number of post-translational modifications such as phosphorylation [31–34], methylation [31, 35], acetylation [36], ubiquitination [31] and SUMOylation [34, 37]. Little is currently known about Sox2 SUMOylation and its function. SUMOylation has been shown to regulate transcription factors' activity through a variety of mechanisms [38].

Sox2 is SUMOylated at lysine 247 *in vivo* and SUMOylation is lost when the target lysine is substituted with an arginine (K247R) [37]. This study showed that endogenous Sox2 was SUMOylated in human gastric cancer cells and that the protein's ability to activate the Fgf4 minimal enhancer in cooperation with the co-activator Oct3/4 was inhibited by SUMOylation. However, it is yet to be determined whether such an effect of Sox2 SUMOylation is general or specific to the Fgf4 enhancer [39]. In fact, SUMOylation of a transcription factor can have the opposite effect, for example it has been shown that SUMOylation of Oct4 increases the protein's stability, DNA binding and transactivation [40, 41]. Wu et al. (2012) showed that SUMOylated Sox2 inhibited the expression of the transcription factor Nanog, while SUMOylated Oct4 increased the expression of Nanog. They proposed a model according to which SUMOylation interferes with the formation of the Oct4-Sox2 heterodimer, which is required to regulate *Nanog*.

Because SUMOylation is known to be involved in the regulation of the activity of other transcription factors, and since Sox2 has been shown to act both as a transcriptional activator and repressor, we set out to test the hypothesis that SUMOylation of Sox2 could regulate the function of Sox2 by altering the balance between its transcriptional activation and transcriptional repression activity. This hypothesis is supported by studies of other Sox factors. For example, SoxE protein could induce neural crest development, while a SoxE SUMO-fusion repressed neural crest formation in *Xenopus* embryos [42]. Lee *et al.* (2012) also showed that SUMOylated SoxE protein loses the ability to bind the co-activator CBP/p300 and recruit the co-repressor Grg4.SUMOylation has also been shown to affect the activity of the SoxB1 transcription factor SoxNeuro in *Drosophila*, which is structurally and functionally homologous to Sox1, Sox2 and Sox3 [43, 44]. SUMOylation of the central nervous system specific SoxB proteins has been conserved during evolution as Sox2 and Sox3 are also targeted by SUMOylation. The mutation of human Sox3 SUMO-targeted lysine to arginine increased Sox3 transcriptional activation activity, suggesting that SUMOylation could inhibit Sox3 transcriptional activation activity [45].

By comparing the effects of over-expressing exogenous wt Sox2, non-SUMOylatable Sox2 mutant or constitutively SUMOylated Sox2 in human neural stem cells (hNSC), we found that SUMOylated Sox2, similarly to wt (SUMOylatable) Sox2, caused the downregulation of a set of proliferation related genes. Many of these genes have been published as Sox2 targets in Chromatin Immunoprecipitation Sequencing (ChIP-Seq) assays. Moreover, we show that SUMOylated Sox2 or wt (SUMOylatable) Sox2 also slow the proliferation of hNSCs. Both of these effects were lost when the SUMO acceptor site of Sox2 was mutated.

## Materials and methods

### Design and cloning of Sox2 mutant constructs

The following Sox2 mutant constructs were constructed. constructedSox2_N48I contains an Asparagine to Isoleucine mutation in position 48, which is in the HMG DNA-binding domain. Sox3 has been shown to be unable to bind the DNA helix when containing an N40I mutation [46–48]. Given the sequence similarity and functional redundancy of Sox2 and Sox3, we hypothesized that the equivalent point mutation in Sox2 would have had the same effect. A sequence alignment revealed that the Asparagine 40 residue present in Sox3 is conserved in Sox2 and corresponds to Asparagine 48. We therefore expected Sox2_N48I not to be able to bind to the DNA. Sox2_NLS is expected not to be translocated into the nucleus due to six amino acids changes (K44T, R45G, R58L, R59G, R115L and R116G) in both of the two presumptive Nuclear Localization Signal sites of the protein [47, 49]. Sox2_HMG-eng contains the Sox2 HMG domain fused to the engrailed repressor domain. This construct is expected to

act as a constitutive transcriptional repressor [50, 51]. Sox2_HMG-VP16 is a fusion protein containing the HMG domain of Sox2 fused to the VP16 activator domain [50]. This construct is expected to behave as a transcriptional activator. Sox2_Δgrg contains mutations in the binding site of Groucho (grg), which is a co-repressor of Sox2, and is expected to prevent Groucho binding [29] by altering four amino acids in positions 203, 207, 208 and 209 (D, L, Q and Y to V, A, A and A). This mutant Sox2 protein has been shown to inhibit the differentiation of NSC to a lesser extent compared to wild-type Sox2 (wt Sox2) [29]. Sox2-SUMO1, Sox2-SUMO2 and Sox2-SUMO3 are fusion proteins containing the wt Sox2 entire protein fused at its C-terminus with SUMO1, SUMO2 or SUMO3 peptides. It has been reported that SUMO conjugation at the target lysine or at the carboxyl terminus of a target protein have a similar effect [37, 52]. The use of C-terminally fused SUMO peptides has been found to mimic the constitutively SUMOylated state of a protein and it has been shown to be a useful approach especially when the SUMOylation site is near the C-terminus of the protein, as it is for Sox2 [42, 52–54]. However, these fusion proteins could potentially be double SUMOylated once transfected into cells as the proteins SUMOylation site (lysine 247) is intact. Sox2_K247A is mutated in the presumptive SUMOylation site [37] and is expected not to be SUMOylated once transfected into cells. In fact, lysine 247 (K247) has been reported to be targeted by SUMOylation [34, 37, 55, 56]. Sox2_K247R is also mutated in the same position of Sox2_K247A and is expected to have lost the ability to be SUMOylated endogenously [37, 56]. Literature reports that mutating lysine residues targeted by SUMO into arginine very often results in a complete or near complete abrogation of the SUMOylation of the target protein [57]. Sox2_K247R-SUMO1, Sox2_K247R-SUMO2 and Sox2_K247R-SUMO3 are fusion proteins containing Sox2_K247R fused at his C-terminus with SUMO1, SUMO2 or SUMO3 peptides. As the SUMO-acceptor lysine is mutated to an arginine, these proteins should not be SUMOylated by the cellular SUMOylation machinery. Therefore, this should prevent the presence of two SUMO peptides, one fused to the C-terminus and the other attached to the SUMO acceptor site, on the protein.

Wt Sox2, Sox2_K247R, Sox2_K247A, Sox2_N48I, Sox2_NLS, Sox2_Δgrg, were obtained through site-directed mutagenesis using QuikChange Lightning Site-Directed Mutagenesis Kit (Agilent Technologies 210518) and QuikChange Lightning Multi-Site Directed Mutagenesis Kit (Agilent 210515) following manufacturer's instructions.

## Western blot

Running gel (12.5% w/v of polyacrylamide) was prepared according to the following recipe, poured between two gel plates in a gel castor and covered with 70% EtOH: 2.14 ml Acryl-Bis 29:1, 1.43 ml 1.57 M Tris HCl pH 8.8 and 0.4% SDS, 1.44 ml water, 75 μl APS 10%, 7.5 μl TEMED. After removal of EtOH a stacking gel (12.5% w/v of polyacrylamide) was prepared according to the following recipe and poured over polymerized running gel: 0.65 ml Acryl-Bis 29:1, 1.25 ml 0.5 M Tris HCl pH 6.8 and 0.4% SDS, 1.73 ml water, 75 μl APS 10%, 7.5 μl TEMED. After solidification the gels were transferred into the gel tank, which was then filled with SDS Running Buffer (30.3g Tris, 188g Glycine, 10ml 10% SDS (in sterile deionized water), sterile deionized water up to 1L).

Sample buffer (Laemmli 2x concentrate, Sigma-Aldrich S3401-10VL) was diluted 1:1 with PBS and added to the frozen cell pellets before running SDS-PAGE. Samples were then boiled at for 10 minutes and incubated on ice for 10 minutes before loading on SDS gel. The protein ladder used for all the experiments of the present study was the SeeBlue Plus2 Pre-Stained Protein Standard (Life Technologies LC5925) and the amount loaded was always 5μl.

The gels were run at 40V until all the samples have reached the running gel, and then at 80V until completion of SDS-PAGE running. The gels were then blotted on Amersham

Protran Premium western blotting membrane (GE Healthcare GE10600118) using the following transfer buffer: 200ml MeOH, 3.03 ml Tris, 14.4 ml Glycine, sterile deionized water up to 1L.

The transfer was run at 30 V overnight. The membrane was then removed and immersed into 5% Marvel Milk in PBS with 0.01% Tween20 (PBST) for at least 1 hour at room temperature, with gentle shaking. The primary antibodies were diluted at the appropriate concentration in PBS added with 0.01% Tween20 (Sigma-Aldrich) (PBST) and used to soak the membrane overnight at 4 °C gently shaking. Following the probing, the membrane was washed in PBST for 5 minutes 3 times while gently shaking, it was then probed with secondary antibody diluted 1/5000 in PBST and incubated for at room temperature for at least 1 hour gently shaking and then washed in PBST for 5 minutes 3 times.

Primary antibodies used: Mouse anti-Sox2 (20G5) Thermo Scientific MA1-014 used (diluted 1:2000), Rabbit anti-HA tag Abcam ab236632 (diluted 1:5000), Mouse anti-Myc tag (9E10) Abcam ab32 (diluted 1:4000). Secondary antibodies used: Donkey anti-mouse IRDye 800CW Li-Cor 926–32212 (diluted 1:5000), Goat anti-rabbit IRDye 680LT Li-Cor 926–68021 (diluted 1:10,000)

## Subcellular fractionation

Buffer A: 10 mM HEPES (pH 7.9), 10 mM KCl, 1.5 mM MgCl2, 0.34 M sucrose, 1 mM dithiothreitol, 10% Glycerol and distilled water. Buffer B: ethylenediaminetetraacetic acid (EDTA) 3mM, ethylene glycol tetra acetic acid (EGTA) 0.2 mM, dithiothreitol 1 mM and distilled water. Protease inhibitor cocktail (Sigma-Aldrich) was added to both buffer A and buffer B just before use. Frozen cell pellets were washed twice with ice cold PBS and centrifuged each time at 1800 xg at 4°C for 2 minutes. Cell pellets were then re-suspended in 100 μl of buffer A with 0.1% of TritonX-100 added (Sigma-Aldrich 11332481001) and incubated on ice for 8 minutes. The samples were then centrifuged at 1300 x g at 4°C for 5 minutes: the supernatants were collected, and the pellets were discarded. The fractions collected were clarified by high-speed centrifugation at 20,000 xg at 4°C for 5 minutes. The supernatants (cytosolic fractions) were collected and placed on ice, while the pellets were washed once with buffer A and lysed by addition of 50 μl of buffer B followed by incubation on ice for 30 minutes. The samples were then centrifuged at 1700 xg at 4°C for 5 minutes and the supernatants (nuclear fractions) were collected and placed on ice while the pellets (chromatin fractions) were washed once with 50 μl of buffer B. All the subcellular fractions were stored at -80°C.

## Culture and transfection of hNSC

ReNcell VM neural progenitor cells (Merck Millipore SCC008) were used in this study as an *in vitro* model of hNSC. They were grown either as neurospheres or as adherent monolayer on laminin coated plasticware in the following culture medium: 50% DMEM:F12 (GIBCO 11554546), 50% Neurobasal (GIBCO 21103049), 0.5% Penicillin-Streptomycin (Sigma-Aldrich P4333-100ML), 1% N2 supplement (GIBCO 17502048), 2% B27 NeuroMix (GIBCO 17504044), heparin (5μg/ml, Sigma-Aldrich 9041-08-1). Human Epidermal Growth Factor (hEGF, GIBCO AF-100-15-1MG) and human Fibroblast Growth Factor (hFGF, GIBCO 100-18B-1MG) were added to the cell culture medium just before use, both at a concentration of 20 ng/ml. Growth medium was replaced with fresh medium every 2 days.

Transfection of hNSC was performed by electroporation using a Nucleofector™ 2b Device (Lonza) with Nucleofector™ Kits for Mouse Neural Stem Cells as per manufacturer instructions (Lonza VPG-1004).

## Luciferase reporter assay

HEK-293 cells were grown in Dulbecco's Modified Eagle Medium (DMEM, Sigma-Aldrich) with 10% foetal bovine serum (FBS, Sigma-Aldrich F2442) and penicillin-streptomycin (Sigma-Aldrich P4333-100ML). Cells were seeded in black 96-well plates and cultured for 24 hours. On the following day, cells were transfected with 25 ng of reporter and 150 ng of activator using Turbofect (Thermo Scientific R0531) and incubated in 5% $CO_2$ for 24 hours. On the following day, culture medium was removed from each well and replaced with 80 μl of 1x Lysis buffer (Dual-Luciferase Reporter Assay System, Promega E1910). Plates were incubated at room temperature for 10 minutes gently rocking and then each sample was pipetted 10 times before transferring the plates to the GloMax luminometer (Promega) for the reading. The luminometer was set to inject 25 μl of Luciferase Reagent for each well with a delay of 2 seconds between injection and measuring.

## RNA sequencing and data analysis

Human NSC were cultured as monolayers in T-75 flasks until 70% confluent, co-transfected with *pcDNA3*, *wt Sox2*, *Sox2_K247R* or *Sox2_K247R-SUMO2* and GFP and seeded in fresh T-75 flasks. The day after transfection, the cells were rinsed with PBS and fresh growth medium added. The following day, cells were harvested and suspended in 660 ml of fresh growth medium. For each sample 160 ml of cell suspension was removed and used for western blot. The remaining 500 ml were run through Beckman Coulter MoFlo XDP to isolate only GFP-expressing cells by Fluorescence-Activated Cell Sorting (FACS). Non-transfected hNSC were used as negative control for cell sorting.

RNA was purified using the RNeasy Plus Mini Kit (QIAGEN 74134) as per manufacturer's instructions. Purified RNA was eluted with 50 ml RNase-free water.

RNA quantity control was performed by Nanodrop (Thermo Scientific TM1000). Quality control was performed using the Agilent 2100 Bioanalyzer System.

The experiment was repeated 3 times, generating 12 samples which were then analysed though RNA sequencing.

RNA samples were sent for RNA sequencing to the Oxford Genomics Centre based at the Wellcome Centre for Human Genetics (Department of Medicine, University of Oxford). They prepared 12 libraries and ran them on 2 lanes of the HiSeq4000 using 75bp PE reads. The libraries were prepared using Illumina's TruSeq Stranded mRNA Library Prep Kit as per manufacturer's specifications with custom primers. Custom primers were used for the PCR enrichment step: Multiplex PCR primer 1.0:

5'–AATGATACGGCGACCACCGAGATCTACACTCTTTCCCTACACGACGCTCTTCCGATCT–3' Index primer:

5'–CAAGCAGAAGACGGCATACGAGAT[INDEX]CAGTGACTGGAGTTCAGACGTGTGCTCTTCCGATCT–3' Indices were designed according to the eight base tags developed by WTCHG (*70*).

Homo_sapiens.GRCh38.dna_sm.primary_assembly.fa reference genome and Homo_sapiens.GRCh38.84.gff3 annotation file were downloaded from:
ftp://ftp.ensembl.org/pub/release-84/fasta/homo_sapiens/dna.

Differential Expression analysis was performed in R using DESeq2.

## Cell proliferation assay

Transfected cells were seeded into two 96-well plates and cultured for 7 days. Medium was replaced every 2 days. Each day 3 replicate wells per transfected sample were analysed using an Alamar Blue cell proliferation assay (Invitrogen DAL1025). One set of wells was used to take

pictures daily and to obtain images showing the cell confluency day by day. The entire experiment was repeated 3 times (biological replicates). Wells were analysed 24 hours after the addition of Alamar Blue.

## Cell death assay

Cell death was detected and measured using CellEvent™ Caspase-3/7 Green Detection Reagent assay (Invitrogen C10423). Human NSC were cultured in a T75 flask for a few passages, harvested and counted before transfection. Cells used for the different samples came from the same population. Cells were then treated with transfection reagent only, cultured with G418 (Sigma G8168-10ML), or transfected with pcDNA3, *wt Sox2*, *Sox2_K247R* or *Sox2_K247R--SUMO2* DNA. Cells from each sample were then used both for the visual assessment experiment and for the first replica of the quantitative experiment (measurement of apoptotic events using a plate reader). After transfection, hNSC were plated both in 6-well plates containing glass coverslips coated with poly-D-lysine and laminin ($4x10^5$ cells/well) for the visual testing experiment and in black 96-well plates with optical clear bottom for the quantitative assessment. All the samples were incubated O/N at 37˚C and the analyses started on the following day (day 1). NSCs cultured O/N with increasing doses of G418 presented increasing amounts of fluorescent cells, confirming that the apoptotic assay can detect apoptotic cells (S1 Fig).

In order to determine how long the samples should be incubated, every set of samples was analysed 30 min, 1 h, 2 h, 24 h and then every 24h until day 5 after addition of CellEvent. The results showed that incubation for 24h showed the highest differences between positive and negative controls (S2 Fig).

## Results

### Sox2 SUMOylation inhibits its transcriptional activation ability

In order to investigate whether SUMOylation affects the transcriptional activity of Sox2, we first tested the ability of Sox2 to activate the expression of a target gene *in vitro* using a *Firefly* luciferase activation reporter assay. In these experiments, the expression of the *Firefly luciferase* gene was regulated by an artificial upstream Sox promoter containing three Sox2 binding sites [29]. The activities of wild-type Sox2 (wt Sox2) and several different Sox2 mutant constructs (S3 Fig) were compared to test whether targeted mutations altered the protein's transcriptional activation function (Fig 1). The activity of the Sox2 constructs was compared to non-transfected cells (UN) and cells transfected with reporter gene only (luciferase). wt Sox2 successfully activated the luciferase reporter gene. As negative controls, we generated and tested a Sox2 mutant construct with a point mutation designed to prevent DNA binding (N48I) as well as a construct mutated at the nuclear localisation site (NLS). As expected, both these constructs were unable to activate the reporter gene. We also generated constitutive repressor (eng) and constitutive activator (VP16) forms of Sox2 by fusing the Sox2 HMG DNA-binding domain to either an engrailed repressor domain or VP16 activator domain. These constructs behaved as expected, with the eng construct lacking activation function and the VP16 construct activating the reporter gene to a similar extent to wt Sox2. Sox2_ΔGrg (a Sox2 construct mutated at the Groucho binding site and therefore incapable of binding to the co-repressor Groucho (35)) did not show any significant difference compared to the positive control. Six different SUMO fusion constructs, which carry SUMO1/2/3 peptides fused to the C-terminus of either wt or the K247R sumo mutant forms of Sox2 and are therefore constitutively SUMOylated (Sox2_-SUMO1/2/3 and Sox2-K247R_SUMO1/2/3), gave a significantly lower signal compared to Sox2-VP16, while two 'SUMO mutants' (Sox2-K247A and Sox2-K247R in which the SUMO-

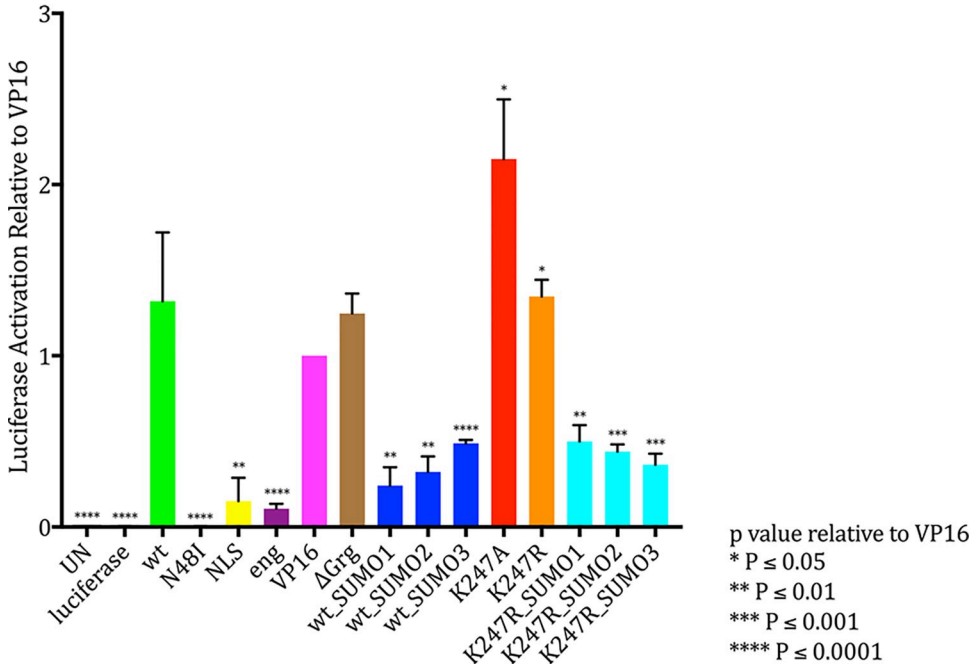

**Fig 1. Luciferase reporter assay performed on HEK-293 over-expressing different Sox2 mutant constructs.**
Average readings of three independent biological replicates together shows that wt Sox2 is statistically different (t test) from the SUMOylation mutant Sox2 constructs (D). Constitutive SUMOylation had an opposite effect compared to the absence of SUMOylation. Data are normalized to the positive control HMG_VP16 (constitutive activator). Statistical differences were calculated using unpaired t test. Bars indicate the standard error of the mean.

acceptor Lysine was mutated) resulted in a significantly higher activation of the reporter gene (Fig 2).

These results suggest that the transcriptional activation ability of Sox2 is reduced *in vitro* when the protein is SUMOylated. This is consistent with results in the literature [37], where a Sox2-SUMO1 fusion construct was shown to lose its ability to activate the *fgf4* minimal enhancer together with the co-factor Oct3/4.

The luciferase reporter assay showed that SUMOylation appears to inhibit the activation activity of Sox2. However, this assay only assessed the ability of Sox2 to activate expression when driven by one particular artificial promoter in HeLa cells and did not assess transcriptional repression. Therefore, the effects of SUMOylation on Sox2 transcriptional activity were further analysed using a genome-wide approach in human hNSCs.

## SUMOylation of Sox2 is necessary for the protein's transcriptional repression activity

Since fusion of Sox2 to all three SUMO subtypes had a similar effect in the luciferase assay described above, we used one of these to study the effect of SUMOylation in more detail. The transcriptional activity of wt Sox2, Sox2_K247R and Sox2_K247R-SUMO2 was investigated by transfection of hNSCs followed by RNA sequencing. Over-expression of the transfected constructs was assessed by western blot (S4 Fig) The expression levels of the exogenous constructs achieved with our transfection protocol was comparable to the levels of endogenous Sox2 (S5 Fig). The experiment was repeated three times (three biological replicates) for each condition (transfection of empty vector, *wt Sox2*, *Sox2_K247R*, *Sox2_K247R-SUMO2*) generating 12 samples, which were analysed by RNA sequencing.

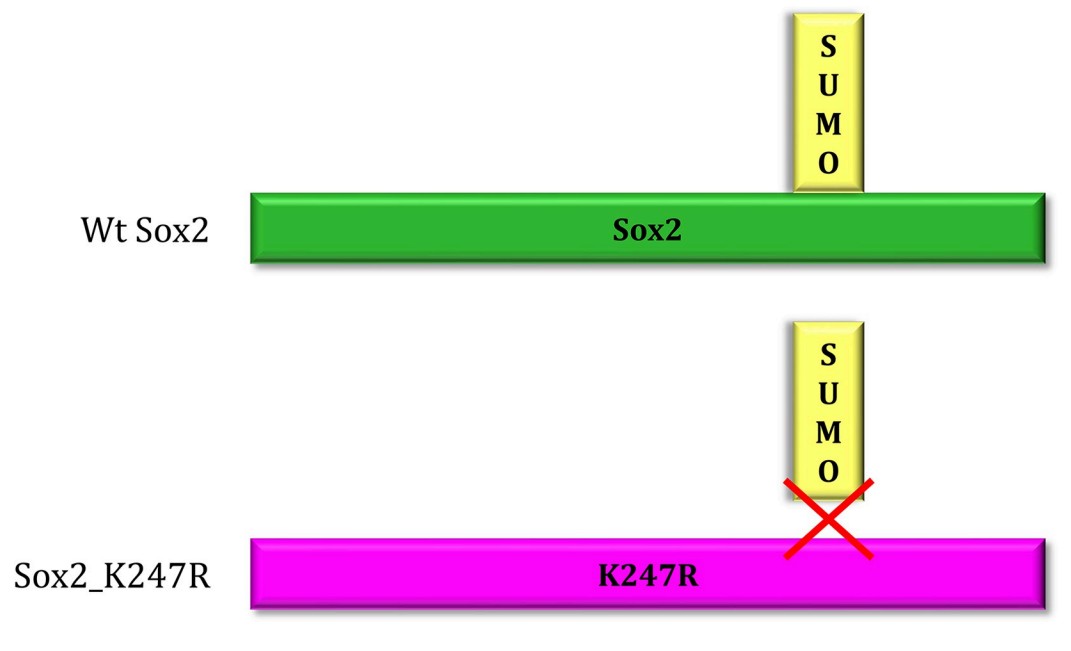

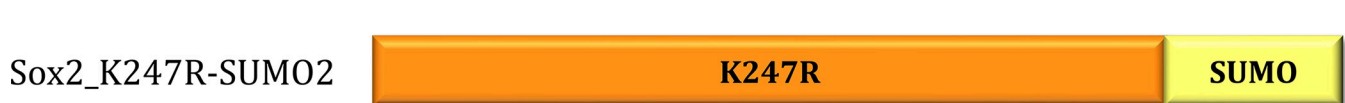

**Fig 2. Schematic representation of Sox2 SUMOylation mutant constructs.** Wt Sox2 can be SUMOylated at K247. The Sox2_K247R construct carries a point mutation that prevents SUMOylation. Conversely, Sox2_K247R-SUMO2 is constitutively SUMOylated at the N-terminus.

Analysis of genes differentially expressed in comparison to control transfected cells (vector only) showed a reduction in the transcriptional activity of Sox2_K247R, the over-expression of which affected only 3.5% of all genes differentially expressed (34 out of 971), compared to wt Sox2 (affected 43% of the genes, 419 out of 971) or the SUMO fusion construct *Sox2_K247R--SUMO2* affecting 74% of all genes differentially expressed (722 out of 971) (Fig 3). Overall, the number of down-regulated genes identified (69%) was higher compared to up-regulated genes (31%). Over-expression of wt Sox2 in hNSCs repressed more genes (320) than it activated (99). Similarly, over-expression of constitutively SUMOylated Sox2 (Sox2_K247R-SUMO2) in hNSCs repressed more genes (503) than it activated (219). On the contrary, over-expression of non-SUMOylatable Sox2 (Sox2_K247R) in hNSCs activated more genes (21) than it repressed (13).

A high proportion of the genes affected by these Sox2 constructs (190 genes, approximately 20% of all differentially expressed genes identified) were common between hNSC over-expressing wt Sox2 and those over-expressing the sumo fusion construct, Sox2_K247R--SUMO2 (3). Most of these genes (25% of all the downregulated genes identified, 165 out of 667) were repressed and none of these shared repressed genes were also repressed in hNSC over-expressing the non-SUMOylatable Sox2_K247R construct (3). This suggests a functional similarity between exogenous wt Sox2, which can be endogenously SUMOylated at the K247 site, and constitutively SUMOylated construct Sox2_K247R-SUMO2 as opposed to non

A

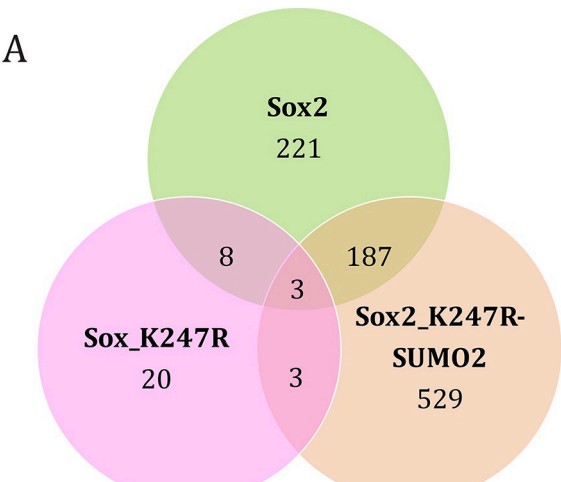

B

Genes Down-regulated

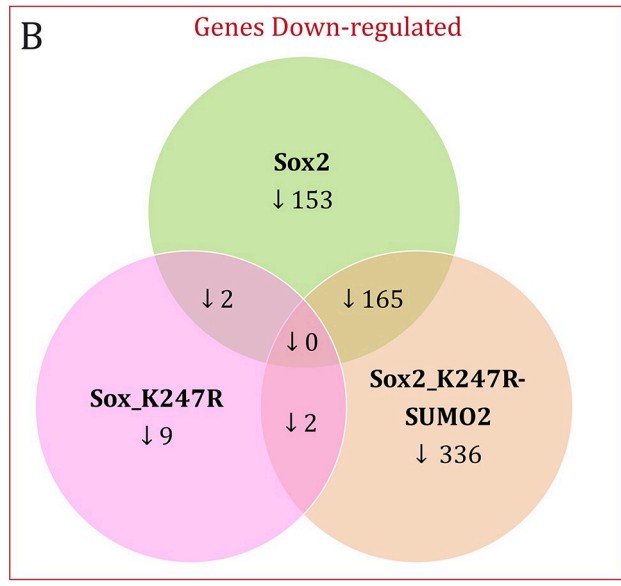

C

Genes Up-regulated

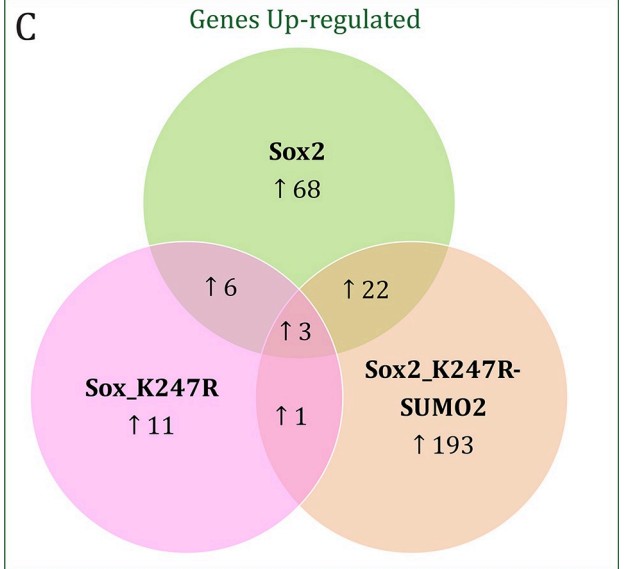

**Fig 3. Comparison of differentially expressed genes in hNSC over-expressing wt Sox2, Sox2_K247R or Sox2_K247R-SUMO2.** Total number of genes differentially expressed (A). Number of genes whose expression was significantly decreased (B) and number of genes whose expression was significantly increased (C).

SUMOylatable Sox2_K247R. In fact, approximately 52% of the genes repressed by wt Sox2 are also repressed by Sox2_K247R-SUMO2 (165 out of 320). Conversely, only 0.6% of the genes repressed by wt Sox2 are also repressed by Sox2_K247R SUMO-deficient construct, suggesting that Sox2 loses its transcriptional repression activity in absence of SUMOylation. Since the Sox2_K247R mutant construct loses the ability to affect the genes that are repressed by wt Sox2, this implies that the effect seen upon expression of exogenous wt Sox2 on these genes is due to that proportion of exogenous Sox2 that becomes SUMOylated. Hence, the reason why the rescued mutant construct (Sox2_K247R-SUMO2) has a stronger transcriptional regulation effect than the wt Sox2, is likely to be because all of the protein produced is now SUMOylated. Together, these data imply that the effects of the expression of these constructs on the genes identified is predominantly due to the SUMOylation of Sox2. This is consistent with the published literature, which reports that, despite a small proportion of SUMO target proteins being actually SUMOylated, it is often the SUMOylated portion of the protein that regulates most of the transcriptional activity of the protein. This phenomenon has been termed the 'SUMO enigma' [58, 59].

The data obtained on repressed genes is in contrast with the data on activated genes, most of which are unique to each construct, with only approximately 22% of the genes activated by wt Sox2 also activated by Sox2_K247R-SUMO2 (22 genes out of 99). This suggests that SUMOylation directly affects Sox2 transcriptional activity causing the downregulation of several target genes.

Since our experiments relied on the function of exogenous Sox2 constructs, we compared the genes affected to those identified as targets of endogenous Sox2 via Chip-Seq analysis. Comparison to the published studies in mouse ESCs and NSCs revealed that 97 of the 165 genes downregulated in our studies were identified as direct Sox2 targets using Chip-Seq analysis. Together with our observation that the levels of exogenous Sox2 appears to be comparable to the level of endogenous Sox2 (S5 Fig), this suggests that, although it's possible that the endogenous Sox2 could affect some non-physiological targets, SUMOylation of endogenous Sox2 is also likely to play a role in its ability to repress target genes.

## SUMO-dependent Sox2 repressed target genes are involved in the regulation of the cell cycle

Gene ontology term enrichment analysis for 'cellular process' (GOrilla, http://cbl- gorilla.cs. technion.ac.il/) performed on the 165 genes down-regulated when either wt Sox2 oror Sox2_K247R-SUMO2 were over-expressed (but not repressed in hNSC over-expressing_K247R) showed an enrichment in genes 'regulating cell cycle process' and 'mitotic cell cycle process' (Table 1). Moreover, gene ontology analysis performed on the same set of genes based on their 'molecular function' revealed an enrichment for 'nucleic acid binding' and 'microtubule binding' functions (Table 2). The same type of analysis performed on the 25 genes up-regulated in hNSC over-expressing wt Sox2 and in hNSC over-expressing Sox2_K247R did not show enrichment of genes involved in any particular cellular process or molecular function.

Protein interaction analysis on the same 165 genes using String software (https://string-db. org/) showed two clusters of genes involved in mitotic cells cycle processes (Fig 4). One of these two clusters (Fig 4, bottom right) included the genes CEP 152 (codifying for

**Table 1. Results of gene ontology analysis of genes down-regulated in hNSC over-expressing wt Sox2 or Sox2_K247R-SUMO2 based on cellular process.**

| P value | Cellular process |
|---|---|
| $< 10^{-9}$ | • Mitotic cell cycle process |
| $10^{-7}$ to $10^{-9}$ | • Regulation of mitotic cell cycle<br>• Cell cycle process<br>• Regulation of mitotic cell cycle phase transition<br>• mRNA splicing<br>• mRNA processing |

Centrosomal Protein 152), CEP250 (codifying for Centrosome-Associated Protein CEP250), CEP290 (codifying for Centrosomal Protein 290), CENPJ (codifying for Centromere Protein J), CNTRL (codifying for 152Centriolin), CDK5RAP2 (codifying for CDK5 Regulatory Subunit Associated Protein 2), AKAP9 (codifying for A-Kinase Anchoring Protein 9) and GOLGA2 (codifying for Golgin A2). All of these genes are involved in mitotic processes including microtubule organisation, centrosome assembly and centromere integrity.

The other gene cluster identified (Fig 4, bottom left) included CENPE (Centrosome-Associated Protein E), CENPF (codifying for Centromere protein F), CLIP1 (codifying for CAP-Gly Domain Containing Linker Protein 1), INCEMP (codifying for Inner Centromere Protein), SMC1A (Structural Maintenance of Chromosomes 1A), SPAG5 (codifying for Sperm Associated Antigen 5), KIF15 (Kinesin Family Member 5) and KIF18b (Kinesin Family Member 18b). These genes are also involved in the formation of the mitotic spindle and in the regulation of mitosis. This suggests that SUMOylation of Sox2 is required for the transcriptional repression of many target genes involved in the regulation of mitosis. In the same comparison with published Chip-Seq data used above, 22 of the 32 cell cycle-related genes downregulated in our studies were identified as direct Sox2 targets. Therefore, the data presented suggest that SUMOylation of Sox2 plays a role in the regulation of hNCS proliferation.

## SUMOylation of Sox2 affects hNSC proliferation

Results obtained in this study suggest that SUMOylation of Sox2 may be important in its ability to regulate hNSC proliferation. Moreover, in all experiments that involved transient over-expression of either *wt Sox2* or *Sox2-SUMO* fusion constructs for two or more days the cell density was noticeably lower than in other samples. Based on this observation, we hypothesised that SUMOylation of Sox2 might be necessary for Sox2 to repress NSC proliferation and that the absence of SUMOylation might ablate this effect. In order to investigate this hypothesis, we set out to compare the proliferation rates of hNSCs over-expressing either pcDNA3 (vector only), *wt Sox2*, *Sox2_K247R* or *Sox2_K247R-SUMO2* (Fig 5). Proliferation rates were measured either 4 hours (S6 Fig) or 24 hours after transfection (Fig 5).

Cells transfected with either pcDNA3, *wt Sox2*, Sox2_K247R or *Sox2_K247R-SUMO2* showed a growth curve that reached its plateau around day 6 after transfection before reaching

**Table 2. Results of gene ontology analysis of genes down-regulated in hNSC over-expressing wt Sox2 or Sox2_K247R-SUMO2 based on molecular function.**

| P value | Molecular function |
|---|---|
| $< 10^{-9}$ | • Nucleic acid binding<br>• RNA binding |
| $10^{-7}$ to $10^{-9}$ | • Tubulin binding<br>• Microtubule binding |

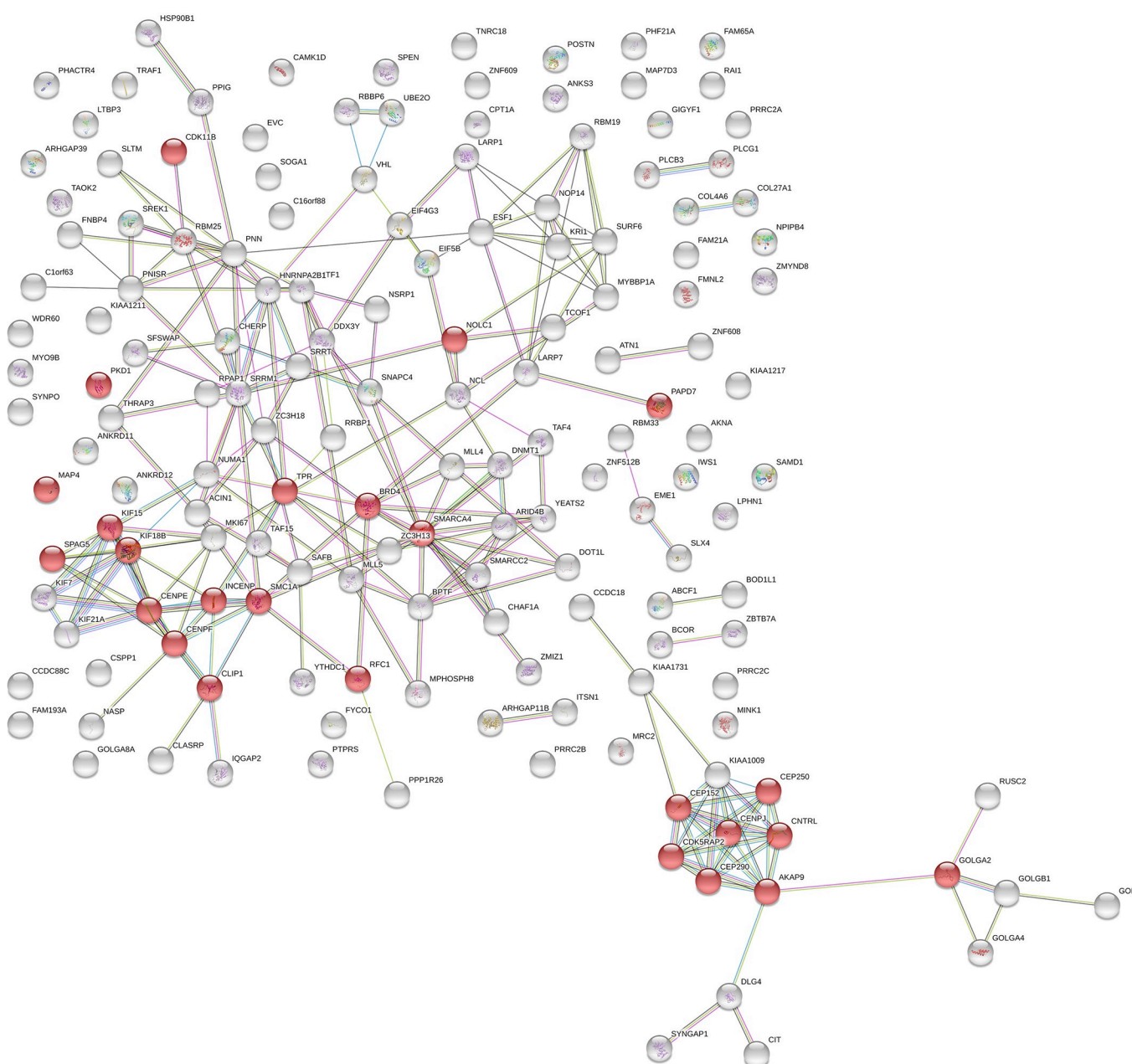

**Fig 4. Gene ontology analysis performed on genes differentially expressed when wt Sox2 and Sox2_K247R are over-expressed.** Genes highlighted in red are involved in cell cycle regulation.

the plateau, samples transfected with either wt *Sox2* or *Sox2_K247R-SUMO2* grew at a slower rate compared to the control (pcDNA3). The rate of growth of cells transfected with *Sox2_K247R* was almost the same as the control.

Statistical comparison between the proliferation rates of the 4 samples showed that the proliferation rate of cells over-expressing either wt Sox2 or Sox2_K247R-SUMO2 was significantly lower than the negative control (pcDNA3) at days 2, 3 and 4 (Fig 6A). These differences tended to disappear 6 days after transfection, presumably due to degradation of the transfected plasmid DNA and its encoded protein and to the cells reaching confluency. At day 7 after

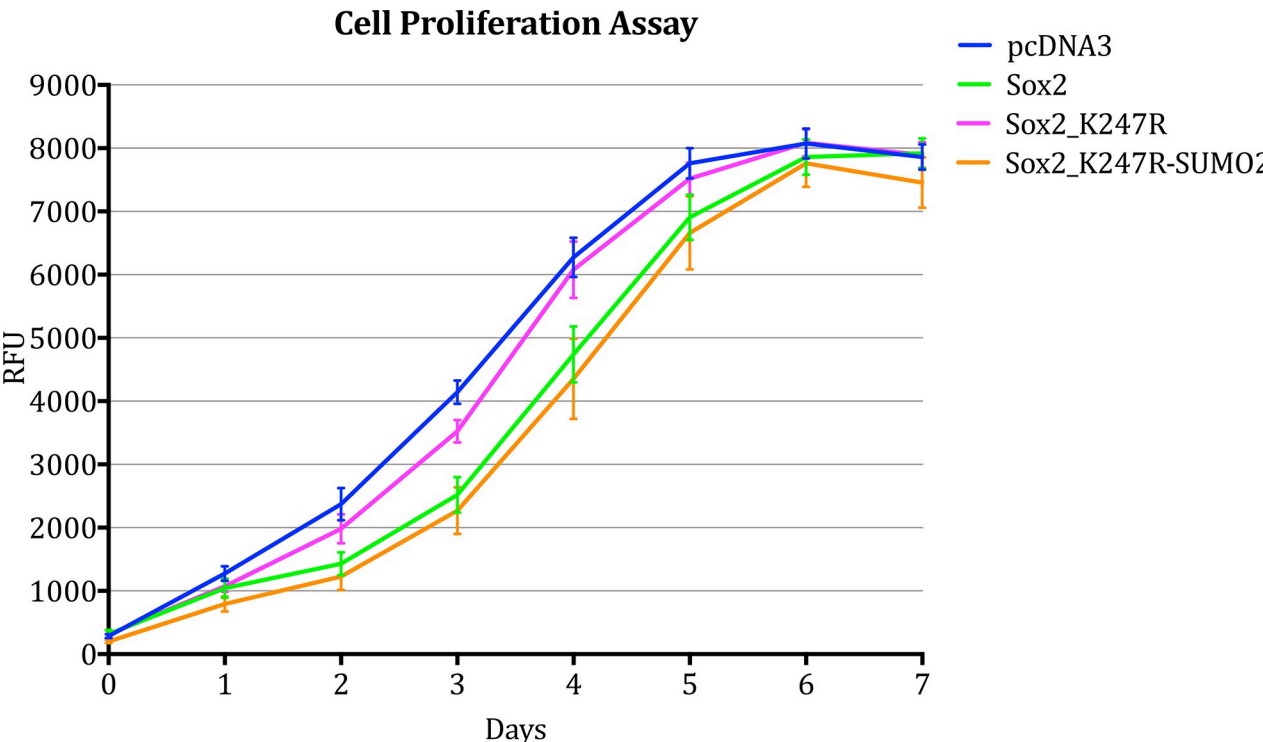

**Fig 5. Proliferation over the course of 7 days of hNSC transiently transfected with either pcDNA3 (empty vector), wt Sox2, Sox2_K247R or Sox2_K247R-SUMO2.** Proliferation rate, measured by Alamar Blue proliferation assay. Bars indicate standard error of the mean. Three biological replicates were performed. Within each biological replicate, three technical replicates were performed.

transfection, there was a slight decrease in the proliferation detected, presumably due to cell death and cell detachment caused by high cell confluency (Fig 6B, panel d7).

Images taken in parallel with the proliferation assays clearly illustrate how cells transfected with *wt Sox2* or *Sox2_K247R-SUMO2* appeared as confluent as cells transfected with *pcDNA3* on day 1 (24 hours after transfection and seeding), but then appeared less confluent for the following days, until approximately day 5 and 6, when they reached a high confluency comparable to that of the control (Fig 6B).

## SUMOylation does not affect apoptosis in hNSC

The data above suggest that SUMOylation of Sox2 could contribute to the regulation of NSC proliferation rate. Such a hypothesis is consistent with the results obtained through RNA sequencing. However, the apparent effect on proliferation could be due to apoptosis induced by over-expression of wt Sox2 or Sox2_K247R-SUMO2. In order to investigate this, we set out to compare the rate of apoptosis in cells transfected with these constructs to samples transfected with *pcDNA3* or *Sox2_K247R*.

Cell death was analysed both on cells grown on microscope coverslips and then visualised after treatment with CellEvent (visual assessment) and on cells grown in a 96-well plate, treated with CellEvent and analysed using a plate reader (quantitative assay).

Samples transfected with either *pcDNA3*, *wt Sox2*, *Sox2_K247R* or *Sox2_K247R-SUMO2* presented very few slightly fluorescent cells, barely visible in the photographic images, with no clear difference in numbers between these samples. By contrast, most positive control cells treated with 10 g/ml G418 were clearly fluorescent (S1 Fig)

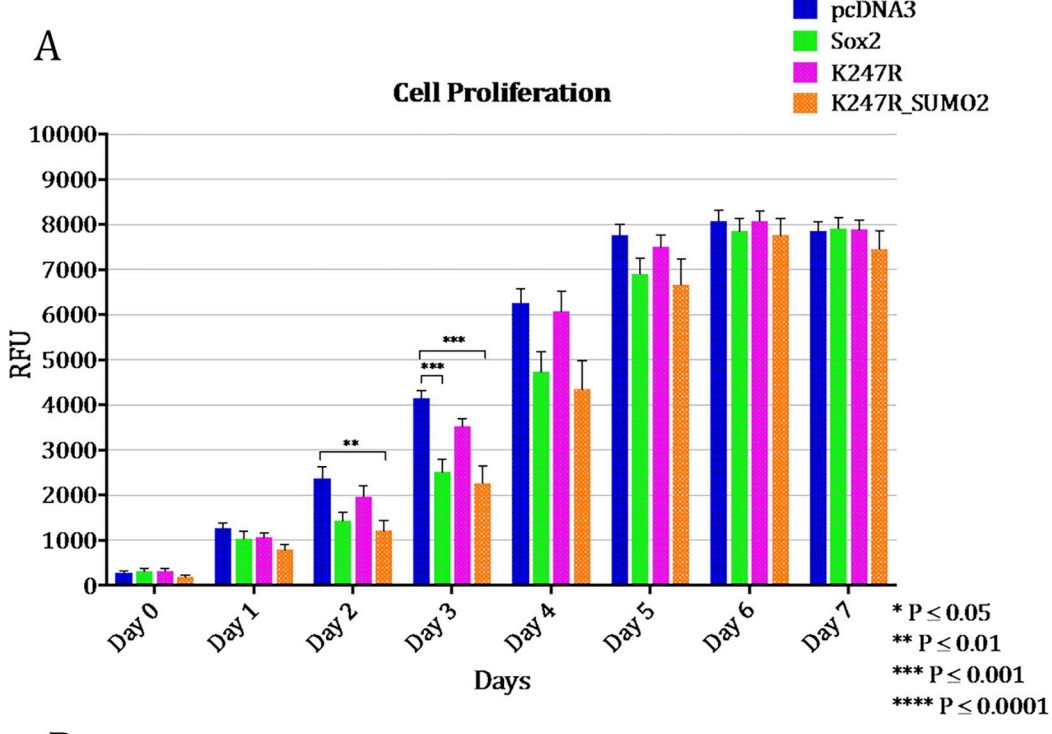

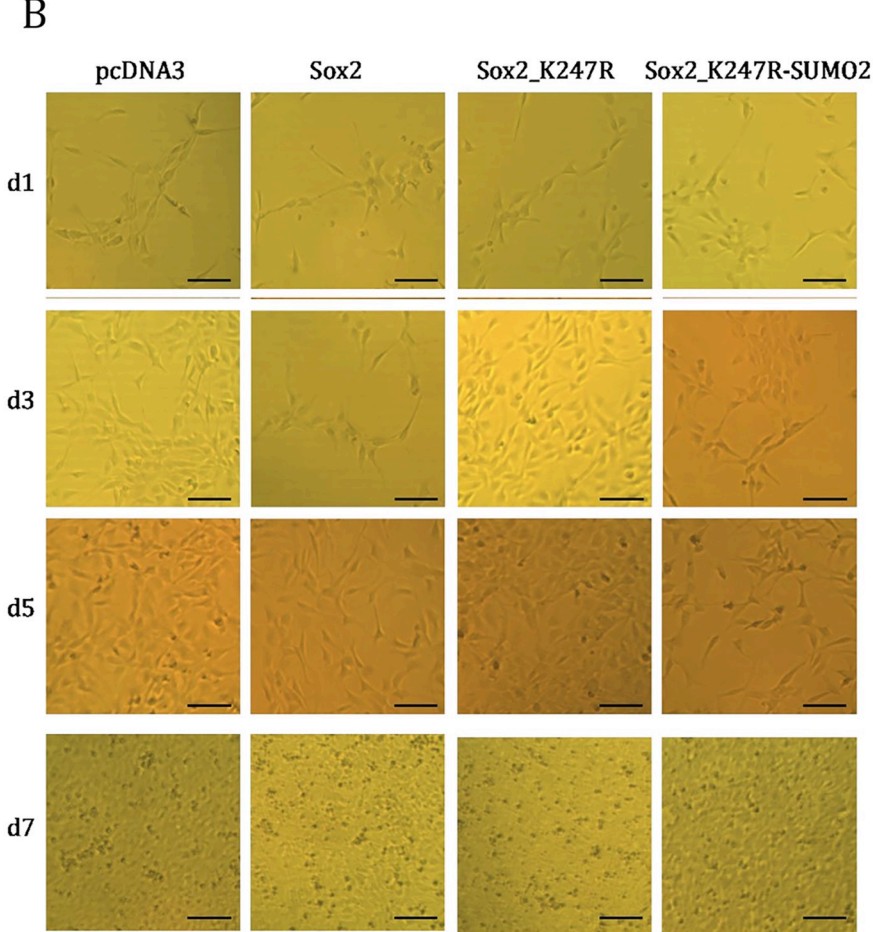

**Fig 6. Comparison between proliferation rates of hNSC transfected with empty plasmid (pcDNA3), wt Sox2, Sox2_K247R or Sox2_K247R-SUMO2 and cultured for 7 days.** (A). Statistical differences were calculated using unpaired t test. Bars indicate standard error of the mean. Three biological replicates (Cell Proliferation 1, 2 and 3) were performed. Within each biological replicate, three technical replicates were performed. Pictures of hNSC over-expressing the different Sox2 constructs and cultured for 7 days (B).

For quantitative analysis of cell death, cells were transfected on day 0 and fluorescence was measured every 24 hours. No significant difference between samples was detected during the course of the experiment, until the faster proliferating samples reached complete confluency, when faster growing cells exhibited increase fluorescence earlier than slower growing samples.

The results of these experiments suggest that transfection of hNSCs with either *wt Sox2* or *Sox2_K247R-SUMO2* affected the cells' proliferation rate rather than affecting their viability (Fig 7).

## SUMOylation does not affect Sox2 subcellular localisation

We next set out to determine whether the effect that SUMOylation of Sox2 has on the protein's transcriptional activation activity could be due to a change in its subcellular localization. Subcellular fractionation of hNSCs over-expressing wt Sox2, Sox2-SUMO2, Sox2_K247R or Sox2_K247R-SUMO2 was followed by western blot of the isolated cellular fractions.

Quantitative analysis by infrared fluorescence imaging confirmed that the wt Sox2 protein was mainly localized in the chromatin and in the cytosol in all cases. Both Sox2_SUMO2 and Sox2_K247R-SUMO2 fusion proteins in the chromatin and cytosol fractions at 3 to 6 fold higher levels than Sox2 and Sox2_K234R, while still presenting a similar overall sub-cellular distribution (Fig 8). This indicates that the SUMO fusion constructs are either more highly

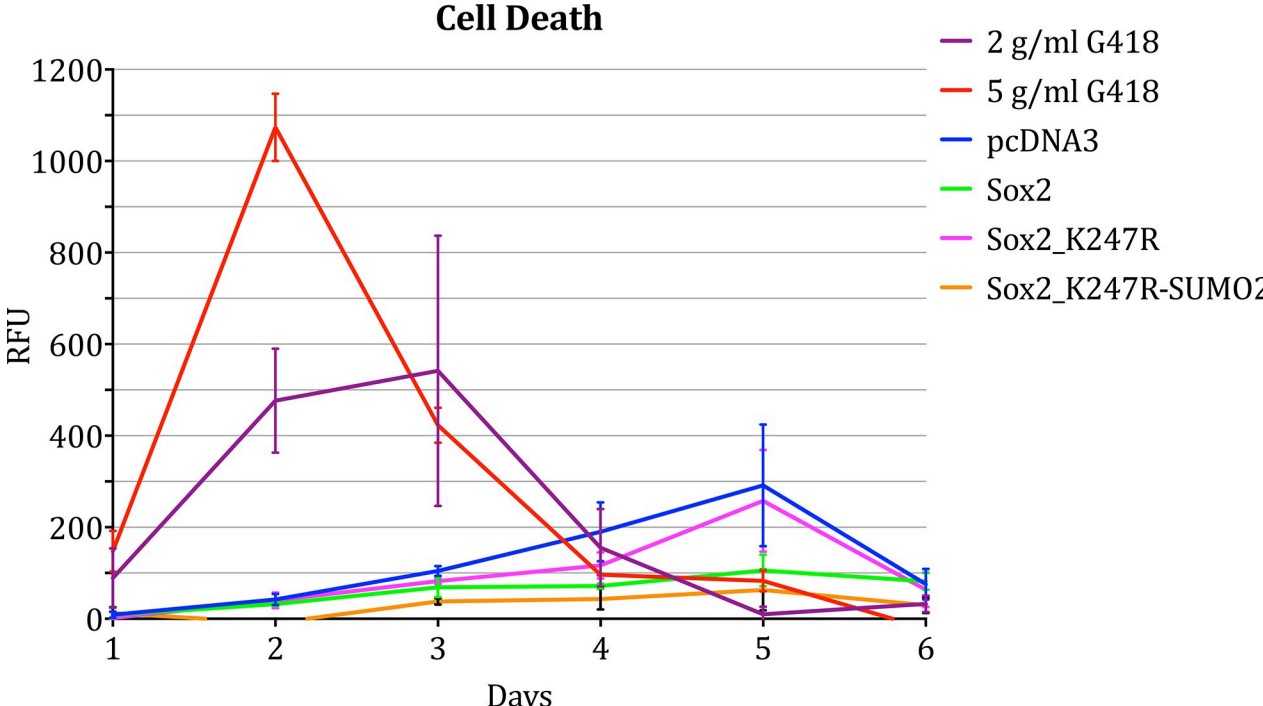

**Fig 7. Cell death of hNSC treated with G418 or transfected with either empty vector (pcDNA3) or different Sox2 constructs.** Apoptosis was measured over the course of 6 days. Two biological replicates of the experiment were performed. Within each biological replicate, two technical replicates were performed. Bars indicate standard deviation of the mean.

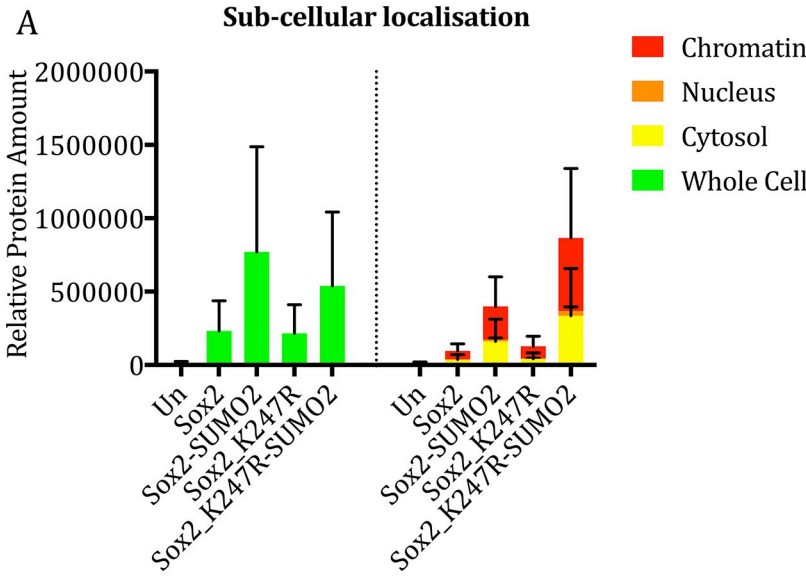

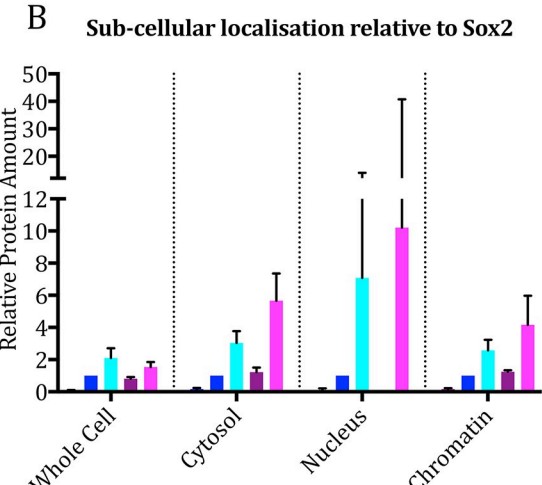

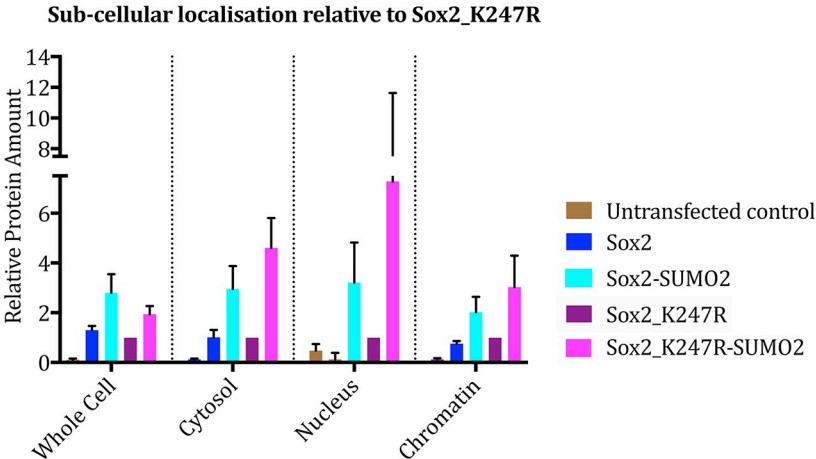

**Fig 8. Quantitative analysis of wt Sox2, Sox2_K247R and Sox2_K247R-SUMO2 across sub-cellular compartments as measured from infrared fluorescence western blot imaging.** (A). The average amounts of proteins in different sub-cellular compartments are shown relative to the amounts of wt Sox2 or relative to the amounts of Sox2_K247R present in each sub-cellular compartment (B). Bars indicate standard error of the mean.

expressed or more stable than the other Sox2 constructs analyzed. Sox2-SUMO2 produced a brighter band compared to wt Sox2 30h after transfection of hNSC (S4 Fig), which could suggest that the Sox2 SUMO2-fusion protein is more stable over time. Thus, although the overall level of protein varied for different constructs, subcellular localization of Sox2 was not affected by SUMOylation.

## Discussion

It is well-established that Sox2 plays a central role in the biology of neural stem cells by maintaining their stemness and regulating their proliferation and differentiation [9, 27, 60]. Our study suggests that SUMOylation of Sox2 is central to its ability to regulate neural stem cell proliferation and that this is largely due to a requirement for SUMO-dependent repression of target genes associated with mitotic cell cycle processes.

Our data suggest that SUMOylation of Sox2 by SUMO2 does not alter Sox2 sub-cellular localisation, consistent with studies reported in the literature [34, 56]. We also found that the level of over-expressed SUMOylated Sox2 in hNSCs was higher than over-expressed wt Sox2. This is consistent with previous western blots performed in our laboratory on hNSC at different times after transfection and suggests that Sox2_SUMO2 might be more stable than wt Sox2 and Sox2_K247R. The hypothesis that SUMOylation could stabilise Sox2 is consistent with reports that SUMO can antagonise ubiquitin and therefore prevent a protein's degradation [61–63]. The results obtained through RNA sequencing are also consistent with this hypothesis as over-expression of Sox2_K247R-SUMO2 affected the expression of more genes (722) than did wt Sox2 (419). This could be explained by the relative stability of each of these constructs: if the Sox2 SUMO2-fusion protein was more stable than the other Sox2 constructs, then its effects on target genes would also be more stable over time, resulting in a higher number of genes being affected at the time when expression levels were analysed, 48 hours after transfection.

Since only a small proportion of Sox2 (probably less than 5%) is normally SUMOylated, the fact that a non-SUMOylatable mutant loses almost all transcriptional activity implies that it is this small fraction of SUMOylated wt Sox2 that is transcriptionally active. If the effect of this mutant was non-specific (not due to loss of SUMOylation) then it would not be rescued by the fusion of SUMO to the mutant's C-terminus. However, it is clear that much of this activity is indeed rescued by addition of a fused SUMO peptide (despite the fact that the construct still carries the same K to R mutation) and this is particularly evident for repressed genes.

Over-expression of wt Sox2 or Sox2_K247R-SUMO2 altered the expression of the same 190 genes, representing approximately 20% of the total number of genes differentially regulated by either of the two constructs. The vast majority of these common genes were repressed (87%). Conversely, out of the genes differentially expressed in cells over-expressing wt Sox2 or Sox2_K247R, only 4% were common and 82% of these were transcriptionally activated by Sox2_K247R. This suggests that the repression of most Sox2 target genes is SUMO-dependent and can be ablated by the lack of the SUMOylation site, K247. This evidence supports the initial hypothesis that SUMOylation of Sox2 affects the protein by enhancing its transcriptional repression activity.

Our results suggest that SUMOylation of Sox2 is essential for its ability to regulate the proliferation rate of hNSCs as, similar to hNSCs over-expressing wt Sox2, cells over-expressing SUMOylated Sox2 grew at a slower rate than controls and down-regulated the same set of mitotic progression regulating genes. Conversely, hNSCs over-expressing Sox2_K247R proliferated at a rate similar to negative control cells and exhibited almost no transcriptional repression activity. This is consistent with Hagey and Muhr (2014) who showed that in mouse embryonic cortical tissue Sox2 represses genes that promote neural precursor cell proliferation. The slow proliferation of NSCs is of upmost importance for their biological role and for preserving the stem cells pool of an organism over time [64]. Therefore, the hypothesis that SUMOylation of Sox2 may be directly involved in the regulation of such a process could have a major impact on the current knowledge of hNSC biology.

Because the activity of Sox2 is tightly linked to the recruitment of co-factors, a possible model that would explain the effect of SUMOylation of Sox2 is that SUMOylated Sox2 could recruit specific co-repressors to repress target cell cycle-related genes. Therefore, if Sox2 loses the ability to be SUMOylated, it also loses the ability to interact with these co-repressors. This model could be tested by protein precipitation purifying Sox2 and its interaction partners, which could then be identified through mass spectrometry.

The importance of the role that Sox2 plays in regulating the pluripotency and the proliferation of NSCs is well known even though the underlying mechanisms of action are still being discovered. It is also known that Sox2 is a versatile transcription factor with the ability of transcriptionally activate and repress different sets of target genes in different cellular and developmental contexts. The mechanisms regulating such functional versatility, while maintaining high target specificity, are also still unclear.

The present study obtained evidence that SUMOylation of Sox2 plays a central role in promoting the function of Sox2 as a transcriptional repressor. This is consistent with the literature, as deficiency of the SUMOylation enzyme, Ubc9, has recently been found to enhance reprogramming of mouse embryonic fibroblasts into induced pluripotent stem cells both *in vivo* and *in vitro* and that SUMOylation functions as a general barrier to cell-fate transitions [65].

We also noted that 34 of the genes differentially regulated by both wt Sox2 and constitutively SUMOylated Sox2_K247R-SUMO2 are involved in human syndromes linked to microcephaly. Of these, 33 are down-regulated by both constructs while one was up-regulated (S1 Table). Since mutations in the closely related SoxB1 gene, Sox3, are associated with human syndromes that include microcephaly, the repression of these target genes might therefore also provide a link between loss of Sox3 activity and microcephaly.

Sox2 has been extensively studied and yet there is much more to uncover on how it is functionally regulated within different cellular contexts. The data shown in the present study open a completely new insight into the molecular mechanisms that regulate the transcriptional activity of one of the major players in stem cell fate determination. These data suggest previously unknown mechanisms that can now be pursued in further studies as it is clear that SUMOylation has a major effect on the regulation of particular Sox2 target genes and consequently on stem cell proliferation.

## Supporting information

**S1 Fig. Visual assessment of apoptosis assay performed on hNSC grown in the presence of G418 (positive controls) or transfected with different Sox2 constructs and cultured O/N.** Scale bars: 50 μm. (TIFF)

**S2 Fig. Measurements of apoptosis on day 1 sets of wells incubated for different amounts of time after addition of CellEvent.** Two biological replicates were performed. Within each biological replicates, two technical replicates were performed. Bars indicate standard deviation of the mean.
(TIFF)

**S3 Fig. Expression of Sox2 constructs in HeLa cells.** Membrane was probed using anti-Myc primary antibody followed by green Licor secondary antibodies.
(TIFF)

**S4 Fig. Western blot performed on hNSC to assess the expression of Sox2 constructs used for RNA Sequencing.** Anti-Myc antibody was used to detect exogenous Sox2 proteins.
(TIFF)

**S5 Fig. Western blot performed on lysates of hNSC transfected with wt Sox2 or Sox_K247R or co-transfected with wt Sox2 or Sox2_K247R, SUMO2 and Ubc9.** All the Sox2 constructs transfected are Myc-tagged and they are the same constructs used thought the present study. The membrane was firstly probed with mouse anti-Myc and rabbit anti-HA primary antibodies and green LiCor anti-mouse and red LiCor anti-rabbit secondary antibodies (A). The same membrane was then probed again using mouse anti-Sox2 primary antibodies and green Licor anti-mouse secondary antibodies (B). The size difference between exogenous and endogenous Sox2 is due to the exogenous Sox2 constructs containing Myc and His tags.
(TIFF)

**S6 Fig. Proliferation of hNSC after transient transfection with either pcDNA3, wt Sox2, Sox2_K247R or Sox2_K247R-SUMO2 and incubation with Alamar Blue for 4 hours.** Bars indicate standard error of the mean.
(TIFF)

**S1 Table. Genes differentially expressed by wt Sox2 and Sox2_K247R-SUMO2 associated with human microcephaly.**
(TIF)

**S1 File. Minimal data sets underlying the results presented and original western blot images.**
(XLSX)

## Author Contributions

**Conceptualization:** Elisa Marelli, Jaime Hughes, Paul J. Scotting.

**Data curation:** Elisa Marelli, Paul J. Scotting.

**Formal analysis:** Elisa Marelli.

**Funding acquisition:** Paul J. Scotting.

**Investigation:** Elisa Marelli, Jaime Hughes.

**Methodology:** Elisa Marelli, Jaime Hughes.

**Project administration:** Paul J. Scotting.

**Resources:** Elisa Marelli, Paul J. Scotting.

**Software:** Elisa Marelli.

**Supervision:** Paul J. Scotting.

**Validation:** Elisa Marelli.

**Visualization:** Elisa Marelli.

**Writing – original draft:** Elisa Marelli.

**Writing – review & editing:** Paul J. Scotting.

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
