## [Decision Letter · Decision Letter 0]

25 May 2023

PONE-D-23-11587SUMO-dependent Transcriptional Repression by Sox2 Inhibits the Proliferation of Neural Stem CellsPLOS ONE

Dear Dr. Marelli,

Thank you for submitting your manuscript to PLOS ONE. After careful consideration, we feel that it has merit but does not fully meet PLOS ONE’s publication criteria as it currently stands. Therefore, we invite you to submit a revised version of the manuscript that addresses the points raised during the review process.

We look forward to receiving your revised manuscript.

Kind regards,

Anujith Kumar

Academic Editor

PLOS ONE

Journal Requirements:

4. PLOS requires an ORCID iD for the corresponding author in Editorial Manager on papers submitted after December 6th, 2016. Please ensure that you have an ORCID iD and that it is validated in Editorial Manager. To do this, go to ‘Update my Information’ (in the upper left-hand corner of the main menu), and click on the Fetch/Validate link next to the ORCID field. This will take you to the ORCID site and allow you to create a new iD or authenticate a pre-existing iD in Editorial Manager. Please see the following video for instructions on linking an ORCID iD to your Editorial Manager account: " ext-link-type="uri" xlink:type="simple">https://www.youtube.com/watch?v=_xcclfuvtxQ"

Additional Editor Comments:

Dear Dr. Marelli,

Thank you for submitting your manuscript to Plos One. I am including the comments that reviewers made on your paper. The referees expressed interest in the study, but they also have a number of criticisms and suggestions. We would be interested in considering a revised version of the manuscript that addresses these concerns in detail.

Reviewers' comments:

Reviewer's Responses to Questions

**Comments to the Author**

1. Is the manuscript technically sound, and do the data support the conclusions?

Reviewer #1: Partly

Reviewer #2: Yes

Reviewer #3: Partly

2. Has the statistical analysis been performed appropriately and rigorously? 

Reviewer #1: I Don't Know

Reviewer #2: Yes

Reviewer #3: I Don't Know

3. Have the authors made all data underlying the findings in their manuscript fully available?

Reviewer #1: No

Reviewer #2: Yes

Reviewer #3: Yes

4. Is the manuscript presented in an intelligible fashion and written in standard English?

Reviewer #1: Yes

Reviewer #2: Yes

Reviewer #3: Yes

5. Review Comments to the Author

Reviewer #1: Manuscript Number: PONE-D-23-11587

Manuscript Title: SUMO-dependent Transcriptional Repression by Sox2 Inhibits the Proliferation of Neural Stem Cells

Marelli et al. describe a study of roles of SUMOylation in the transcriptional activity of Sox2 in a human cell line of early neural progenitors. By transfecting a variety of cDNA constructs, including wild type Sox2, or a mutant Sox2 unable to be SUMOylated , or the same mutant “rescued” by C-Terminal addition of SUMO peptides, they demonstrate the ability of the rescued SUMOylated mutant to affect gene transcription in the transfected cell line, by increasing or repressing the activities of almost a thousand genes. The non-SUMOylatable Sox2 cDNA is instead essentially devoid of such activity. Within the subset of repressed genes, they point to a significant enrichment of genes related to cell cycle, and suggest that the inhibition of these genes may be relevant to the control of the progenitor proliferation kinetics and their differentiation. The identification of these cell-cycle related genes as target of repression by Sox2 is potentially important.

Major Comment 1

The experiments are, in general, performed in a technically sound way, with appropriate controls and repetitions adequate for statistical analysis, and the data are therefore credible. However, I have some reservations on the strategy used by the authors, and on the interpretation of the results. In Figure 1, they show that a non-SUMOylatable mutant is more efficient than a wt-Sox2 or “rescued” non-SUMOylatable Sox2 mutant in driving transcription of a Luciferase construct having a Sox2-binding sequence in the promoter. Thus, lack of SUMOylation does not seem to impair the transcriptional activity of Sox2.

As this type of experiment does not allow to test for repression, they try a different strategy, that yields results in contrast with those described above, suggesting that Sox2 SUMOylation matters, both in activation and in repression of transcription. Unfortunately, the authors do not discuss this discrepancy and do not discuss their results in sufficient depth.

This experiment is unusual, and rather difficult to interpret. The authors do not provide some important details, and do not describe the logic of the experiment, and possible caveats, in depth. To evaluate the role of Sox2 SUMOylation they transfect a wild type Sox2 cDNA construct, a mutant non-SUMOylatable Sox2 construct, and a rescued mutant non-SUMOylatable Sox2 carrying SUMOpeptides at its C-terminal (from now-on, wt Sox2, mut Sox2 and rescued mut Sox2, respectively).The recipient cell line is a human neural progenitor line. Importantly, neural progenitor lines typically express Sox2 at relatively high levels, meaning that gene expression in these cells reflects the action of Sox2, that controls a significant proportion of the genes. It is not clear to me how much Sox2 is expressed in these cells. If Sox2 is indeed present in these cells, significantly modifying the expression of genes regulated by ENDOGENOUS Sox2 may require the addition of large amounts of EXOGENOUS Sox2, in quantities that may well exceed physiological levels. In their transfection experiments (Fig.3), the authors show that both wt Sox2 and the rescued mut Sox2

(Sox2_K247R_SUMO2) are able to up- and down-regulate a small proportion of the expressed genes. In contrast, non-SUMOylatable mut Sox2 (Sox2_K247R9) is essentially devoid of both activating and repressing activity. The authors comment (lines 332-333):” This evidence clearly shows that SUMOylation of Sox2 has a strong repressive effect on target genes and that this effect is rescued by the absence of SUMOylation”. I do not agree with this statement, for two reasons: First, a similar effect, although somewhat less strong, is also obtained with wt Sox2. It is true that wt Sox2 can be SUMOylated, after being newly synthesized following cDNA transfection, but the authors report that only about 5% of Sox2 is SUMOylated within the cell, see lines 507-508. So, it remains to be proven that non-SUMOylated Sox2 is inactive in gene repression.

The second objection is that it is logically incorrect to equate the proportion of genes downregulated by Sox2 overexpression to the absolute activity of Sox2 expressed at physiological levels. Genes that are already downregulated by endogenous factors, including Sox2 itself, may well require much higher levels (not necessarily physiological) of Sox2 and other repressing factors to be further repressed. So, the different results obtained using mut Sox2 (Sox2_K247R9) versus rescued mut Sox2 (Sox2_K247R_SUMO2) might simply indicate that mutSox2 is somewhat less efficient in inducing a high level of repression than rescued mut Sox2, and fails to reach the high expression threshold necessary for further repression of endogenously repressed genes. This does not imply that the basic repression activity of Sox2 is altered or abolished.Note that, at lines 474-475 the authors report that mutSox2 is much less present in chromatin than rescued mut Sox2, which might contribute to the inability of mut Sox2 to further repress the expression of genes expressed in the transfected cells. These results, overall, do not imply that non-SUMOylated Sox2 is unable to repress Sox2 targets.

As to activation by SUMOylated versus non-SUMOylated Sox2, the authors suggest, again based on data in Fig.3, that only SUMOylated Sox2 is transcriptionally active (see lines 507-508). The reasoning reported above as to repression data, can be applied equally well to the activation data. Moreover, Fig.1 clearly shows that, when tested on the same reporter gene, carrying Sox2-binding sites, constitutively SUMOylated Sox2 is LESS active than wt Sox2 or non SUMOylatable Sox2

Major Comment 2

The genes analysed in Fig.3 represent the complete transcriptome of the tested cells. A proportion of these genes are presumably directly regulated by Sox2; in addition, genes regulated by Sox2 may encode factors which in turn affect the expression of genes non directly controlled by Sox2 ( for a knock-out experiment in mouse neural cells reporting expression data, see Bertolini J et al., 2019, Cell Stem Cell, PMID: 30849367). The authors should carefully consider these points in Discussion. Do transfected Sox2 constructs affect, in their experiments, only some of the physiological targets of Sox2? Overexpression experiments have the drawback that genes not regulated, in vivo, by the transcription factor under study, or only moderately affected by it, may be inappropriately activated; this effect is well known in molecular oncology. So the question is whether in their experiments the transfected constructs encoding Sox2 and its SUMOylatable or non SUMOylatable variants, activate/repress the real Sox2 targets. The authors might consult the data in Bertolini et al (see above) and ChIP-seqs for Sox2 available in the literature; see also D’Aurizio R.et al., 2022Int.J.Mol.Sci., PMID:35887306, who compare mouse with human data and provide many references to papers regarding ChIP-seqs).Defining which ones, among the repressed cell cycle-related genes, are indeed regulated in vivo by Sox2 would be very important.

Minor Comments

1)In the cell proliferation experiment of Fig.5, differences are initially detected between day 1 and 2, and slightly increase at day 3; then, curves increase in parallel for 2-3 days, when the curves start approaching a plateau. So, the effects of the transfected constructs seem to be only at the initial stages, then they progressively disappear.

2)lines 353-358

See major comment 1. Gene downregulation is observed with transfection of both wtSox2 and rescued mut Sox2 (Sox2_K247R_SUMO2.

3)lines507-509

I strongly disagree with this comment, see Major Comment 1

4)lines 516-520

“our hypothesis that 517 Sox2 SUMOylation regulates the protein activity as transcription factor” : this sentence is unclear.

“we showed that SUMOylated Sox2 acquires transcriptional 520 repression activity compared to non SUMOylatable mutant Sox2”: not true, see comment 1

5)lines 551-552 “Our data suggest that when Sox2 cannot be SUMOylated, it loses its ability to repress a set of cell-cycle regulating target 552 genes, resulting in a higher proliferation rate”. See comments 1 and 2. Also, the higher proliferation rate is very transient. The experiment shoul better be performed in long-term growing cell cultures.

6)It would important to have access to Tables listing the top 50 most upregulated and the top 50 most downregulated genes in the transfection experiments reported in Fig.3, together with their expression values, as Transcript per Million, and fold changes.

7) English spelling and some sentences should be revised, see for example “which we have showed affects (lines 383-4), “grew slower” (line 416), wells was analyzed (line 254), underling (line 561),and several other ones.

CONCLUSIONS

This paper presents some interesting results, in particular the observation of the downregulation of many genes, in particular genes involved in cell cycle control. If the authors can confirm that these downregulated genes are physiological Sox2 targets, shown to harbour Sox2 -binding sites, this is of great potential interest. As to the role of SUMOylation, at present the data are not entirely convincing, in my opinion. The ideal experiment would be to introduce a mutation within the SUMOylation site of Sox2 by CRISPR-CAS technology, and to select appropriate cell lines. I have to recognize, however, that this is a complex experiment. Alternatively, once a bona fide Sox2 target gene is identified among the repressed genes, one might transfect an appropriate construct into Sox2-null cells, and then transfect these cells with SUMOylated (or SUMOylatable) and non-SUMOylatable Sox2 constructs to study the relative effects of these constructs on this target.

Reviewer #2: In the present work, the authors studied the role of Sox2 SUMOylation in human neural stem cells (NSCs). They generated Sox2 mutants with diminished SUMOylatability and mutants with a SUMO fusion (constitutively SUMOylated) and studied their transcriptional activity on a reporter system and their effect on NSCs’ transcriptome and proliferation. They found that the variants fused to SUMO lost their transcriptional activation activity compared to the wt Sox2 and acquired transcriptional repression activity compared to non SUMOylatable Sox2 mutants. They identified a set of genes that were differentially regulated by the different Sox2 variants in NSCs. Interestingly, the genes repressed by Sox2 in a SUMO-dependent manner are associated with cell cycle control. In agreement, they also found that SUMOylation of Sox2 affected the proliferation of these cells, since the NSCs transfected with wt Sox2 or Sox2 fused to SUMO grew slower than control cells or that cells transfected with the Sox2 mutant with impaired SUMOylation. Since they did not detect differences in cell viability, they conclude that the effect is on the proliferation rate. Finally, the authors studied the subcellular localization of the different mutants through western blot analysis of subcellular fractions. Although they did not find any differences in their localization, they detected differences in the expression of the different mutants and proposed that SUMOylation may stabilize Sox2, but they suggest that this may not be the only role played by SUMOylation in the regulation of Sox2 activity. Finally, they commented that various genes differentially regulated by both wt Sox2 and the SUMO fusion are involved in human syndromes linked to microcephaly and suggested a link between the loss of activity of Sox3, a member of Sox family, and microcephaly.

Major points:

-Interference with the SUMOylation pathway, for example by transfection of a dominant negative mutant of the UBC9 enzyme, at least in the reporter assay, would strengthen the conclusions.

-The authors could complement the localization analysis with a microscopy approach, ideally in living cells expressing fusions of the different mutants to a fluorescent protein; or alternatively, by immunofluorescence of fixed cells (they use an Anti-Myc antibody to detect by Western blot the expressed constructions that have the Myc tag).

Minor points:

-The authors select transfected cells by FACS for the RNA-seq analysis; however, I could not find any description of selection or identification of transfected cells in the proliferation analysis. How much of the cell population is transfected? They mention that the differences observed “tended to disappear around 5-6 days after transfection, presumably due to degradation of the plasmid DNA transfected and its encoded protein.” In my opinion, identification of transfected cells (for example by expressing the Sox2 variants fused to a fluorescent protein) would strengthen the conclusions.

- Analysis of the stability of the different Sox2 variants would contribute to the discussion about the role played by SUMOylation in the regulation of Sox2 activity.

-At the end of the introduction section, a paragraph with the most relevant findings of the manuscript is missing.

-Tables 1 and 2 are the same.

- In my opinion, data from figures 5 and 6 should be included in one figure, it is not necessary to show the three replicates in the way that are presented in the main figure 6A, and scale bars should be included in Figure 6B.

-Line 254: Wells was analyzed should be replaced by Wells were analyzed.

-Line 453: fester should be replaced by faster.

-Line 487: maintaining their stemness state should be replaced by maintaining their stemness.

Reviewer #3: In this study, Marelli et al investigated how SOX2, SOX2 mutant deficient of SUMOylation, and SOX2-SUMO fusion proteins regulate transcription and cell proliferation in neural stem cells. They observed that SOX2 represses more genes than it activates. The genes repressed by SOX2 are associated with centrosomes, centromeres and other aspects of cell cycle control. In addition, they observed that the SOX2-SUMO fusion protein is more potent in repression as well as activation than wildtype SOX2, whereas the SUMOylation-defective mutant is poorly active in regulation of gene expression. Thus, they conclude that SUMOylation of Sox2 is necessary for its repression of gene expression and for its repressive effects on cell proliferation. However, there are a nubmer of issues needed to be addressed before I will support its publication.

1. In Fig. 1, authors showed that K247A and K247R mutants were as active as SOX2 in transcriptional activation of a reporter gene. The Western blot data for expression of various SOX2 mutants should be showed in this figure to interpret the luciferase reporter assay.

2. Since K247A and K247R were highly active in assay in Fig. 1, why it was essnetially inactive in transcriptional regulation (only repressed 19 and activated 11 genes) in neural stem cells?

3. A major concern for the study is that RNA-seq was performed for each type of SOX2 proteins for only one dose. Much of the differences in regulation of gene expression could be attributed to different levels of SOX2 expression. For instance, K247R mutant could be expressed very low in this experiment, and thus had little effect on gene expression. It would be more convincing if two or three differnt doses of SOX2 were tested for each and data were compared among the samples that have similar level of SOX2 proteins.

4. Authors need to address if SUMOylation influences SOX2 transcriptional activity primarily by affecting SOX2 protein stability or promoting its repression activity or both.

5. Introduction is too long and some of information could be deleted or moved to discussion.

6. PLOS authors have the option to publish the peer review history of their article (what does this mean?). If published, this will include your full peer review and any attached files.

Reviewer #1: **Yes: **SERGIO OTTOLENGHI

Reviewer #2: No

Reviewer #3: No

---

## [Author Response · Author response to Decision Letter 0]

11 Dec 2023

Dear PlosONE editorial team,

Although, we are now no longer in a position to carry out more experiments since the Scotting lab was closed due to Dr. Scotting’s retirement, we have addressed all of the reviewers’ comments in our response below and in the edits that we have applied to the paper. 

Reviewer 1, major comment 1: “The experiments are, in general, performed in a technically sound way, with appropriate controls and repetitions adequate for statistical analysis, and the data are therefore credible. However, I have some reservations on the strategy used by the authors, and on the interpretation of the results. In Figure 1, they show that a non-SUMOylatable mutant is more efficient than a wt-Sox2 or “rescued” non-SUMOylatable Sox2 mutant in driving transcription of a Luciferase construct having a Sox2-binding sequence in the promoter. Thus, lack of SUMOylation does not seem to impair the transcriptional activity of Sox2. As this type of experiment does not allow to test for repression, they try a different strategy, that yields results in contrast with those described above, suggesting that Sox2 SUMOylation matters, both in activation and in repression of transcription. Unfortunately, the authors do not discuss this discrepancy and do not discuss their results in sufficient depth.”

The data shown in figure 1 suggests that by blocking Sox2 SUMOylation, its transcriptional activation ability is increased compared to wild-type (wt) Sox2. Conversely, constitutive SUMOylation of Sox2 has the opposite effect of diminishing the protein’s ability to transcriptionally activate the reporter gene. This assay is not representative of an in vivo system, as the promoter controlling the expression of the Luciferase gene has been artificially designed by inserting three repetitions of a Sox2 binding motif in a generic promoter (pTl/3xSX promoter, as published by Liu et al. [1]). In vivo, Sox2 binding consensus motif can vary [2] and Sox2 can regulate target genes both directly and indirectly [3, 4]. Therefore, as stated in the manuscript’s results section, this assay only allowed us to detect and measure direct transcriptional activation of the artificial promoter and was used as a screening method to test several Sox2 mutant constructs to assess if the mutations applied impacted the protein’s transcriptional activity compared to wt Sox2. In order to make this clearer, we have added the following to the paper: “The luciferase Reporter Assay showed that SUMOylation appears to inhibit the activation activity of Sox2. However, this assay only assessed the ability of Sox2 to activate expression driven by one particular artificial promoter in HeLa cells and did not assess transcriptional repression. Therefore, the effects of SUMOylation on Sox2 transcriptional activity were further analysed using a genome-wide approach in hNSCs” (lines 350-355).

Moreover, while our study focussed mainly on the effects of Sox2 SUMOylation on its transcriptional repression ability, we do not intend to imply that Sox2 transcriptional activation ability is not also affected by SUMOylation. In fact, the RNA sequencing data shown in figure 3 suggests that this is also the case. We decided to focus our study on the effect on transcriptional repression for the following reasons. Many genes repressed by exogenous wt Sox2 and SUMOylated sox2 were common to both constructs while gene activated were not; the effect of loss of SUMOylation of Sox2 affected repressed genes more strongly than activated genes; only those genes repressed fell into functional groups and many of these were cell cycle related, which has a direct link with the effect of exogenous Sox2 on cell proliferation. 

Reviewer 1, major comment 1: “To evaluate the role of Sox2 SUMOylation they transfect a wild type Sox2 cDNA construct, a mutant non-SUMOylatable Sox2 construct, and a rescued mutant non-SUMOylatable Sox2 carrying SUMO peptides at its C-terminal (from now-on, wt Sox2, mut Sox2 and rescued mut Sox2, respectively). The recipient cell line is a human neural progenitor line. Importantly, neural progenitor lines typically express Sox2 at relatively high levels, meaning that gene expression in these cells reflects the action of Sox2, that controls a significant proportion of the genes. It is not clear to me how much Sox2 is expressed in these cells. If Sox2 is indeed present in these cells, significantly modifying the expression of genes regulated by ENDOGENOUS Sox2 may require the addition of large amounts of EXOGENOUS Sox2, in quantities that may well exceed physiological levels.”

We acknowledge that levels of expressions of endogenous and exogenous Sox2 might play an important role in the interpretation of our RNA sequencing data. Therefore, we have added further data in the supplementary (Supplementary figure S3): the western blot below, which was performed for a different purpose, was initially probed with anti-Myc primary antibody (exogenous Sox2 is Myc tagged) and subsequently with anti-Sox2 primary antibody. The blot shows bands of comparable brightness between endogenous and exogenous Sox2. Although these are different primary antibodies and might have slightly different avidity, we used the same fluorescein-tagged anti-mouse secondary antibody to visualize them. We believe it is reasonable to conclude that the levels of exogenous Sox2 are not dissimilar to the levels of endogenous Sox2. Therefore, we have added the following statement in the manuscript: “The expression levels of the exogenous constructs achieved with our transfection protocol was comparable to the levels of endogenous Sox2 (S5 Fig)” (lines 376-378. This is consistent with the new data we present below, showing that many of the Sox2 targets we have identified, have also been identified through ChIP-Seq analysis in other published studies. Reviewer 1, major comment 1: “In their transfection experiments (Fig.3), the authors show that both wt Sox2 and the rescued mut Sox2 (Sox2_K247R_SUMO2) are able to up- and down-regulate a small proportion of the expressed genes. In contrast, non-SUMOylatable mut Sox2 (Sox2_K247R9) is essentially devoid of both activating and repressing activity. The authors comment (lines 332-333):” This evidence clearly shows that SUMOylation of Sox2 has a strong repressive effect on target genes and that this effect is rescued by the absence of SUMOylation”. I do not agree with this statement, for two reasons: First, a similar effect, although somewhat less strong, is also obtained with wt Sox2. It is true that wt Sox2 can be SUMOylated, after being newly synthesized following cDNA transfection, but the authors report that only about 5% of Sox2 is SUMOylated within the cell, see lines 507-508. So, it remains to be proven that non-SUMOylated Sox2 is inactive in gene repression.”

Since the Sox2_K247R mutant construct loses the ability to affect the genes that are repressed by wt Sox2, this implies that the effect of exogenous wt Sox2 on these genes is due to that proportion of exogenous Sox2 that becomes SUMOylated. Hence, the reason why the rescued mutant construct (Sox2_K247R-SUMO2) has a stronger transcriptional regulation effect than the wt, is because 100% of the protein produced is now SUMOylated. Together, these data imply that the effects of the expression of these constructs on the genes identified is predominantly due to the SUMOylation of Sox2.

If non-SUMOylated Sox2 were in general active as a repressor, we would expect a much greater number of the identified genes to be repressed by exogenous, non-SUMOylatable Sox2 (Sox2_K247R). 

The hypothesis that the portion of exogenous wt Sox2 that becomes SUMOylated is responsible for the transcriptional regulation of a high number of genes, is consistent with the published literature. Yau et al. published that: “SUMOylation targets are rarely quantitatively SUMOylated. In fact, in the case of most targets, only a small percentage of a SUMO target protein is SUMOylated at any given time. This is paradoxical because, as will become clear, SUMOylation often modulates the activity of nearly the entire population of a protein target. This phenomenon has been termed the ‘SUMO enigma’” [5, 6]. 

In order to clarify this point, we have added the following: 

“Since the Sox2_K247R mutant construct loses the ability to affect the genes that are repressed by wt Sox2, this implies that the effect seen upon expression of exogenous wt Sox2 on these genes is due to that proportion of exogenous Sox2 that becomes SUMOylated. Hence, the reason why the rescued mutant construct (Sox2_K247R-SUMO2) has a stronger transcriptional regulation effect than the wt Sox2, is likely to be because all of the protein produced is now SUMOylated. Together, these data imply that the effects of the expression of these constructs on the genes identified is predominantly due to the SUMOylation of Sox2. This is consistent with the published literature which reports that, despite a small proportion of SUMO target proteins being actually SUMOylated, it is often that SUMOylated proportion of the protein that regulates most of the transcriptional activity of the protein. This phenomenon has been termed the ‘SUMO enigma’ [5, 6]” (lines 404-414).

Reviewer 1, major comment 1: “The second objection is that it is logically incorrect to equate the proportion of genes downregulated by Sox2 overexpression to the absolute activity of Sox2 expressed at physiological levels. Genes that are already downregulated by endogenous factors, including Sox2 itself, may well require much higher levels (not necessarily physiological) of Sox2 and other repressing factors to be further repressed. So, the different results obtained using mut Sox2 (Sox2_K247R9) versus rescued mut Sox2 (Sox2_K247R_SUMO2) might simply indicate that mutSox2 is somewhat less efficient in inducing a high level of repression than rescued mut Sox2 and fails to reach the high expression threshold necessary for further repression of endogenously repressed genes. This does not imply that the basic repression activity of Sox2 is altered or abolished. Note that, at lines 474-475 the authors report that mutSox2 is much less present in chromatin than rescued mut Sox2, which might contribute to the inability of mut Sox2 to further repress the expression of genes expressed in the transfected cells. These results, overall, do not imply that non-SUMOylated Sox2 is unable to repress Sox2 targets.”

Although we accept that we cannot make any certain predictions of the role of SUMOylation of the endogenous Sox2, we would argue that the strength of the effect caused by loss of SUMOylation of the exogenous Sox2, suggests that SUMOylation of endogenous Sox2 is likely to have a substantial effect on its ability to repress target genes. So, in the paper, we have now re-phrased our statement for the interpretation of the data. Rather than “This evidence clearly shows that SUMOylation of Sox2 has a strong repressive effect on target genes and that this effect is rescued by the absence of SUMOylation”, we now say: “Since our experiments relied on the function of exogenous Sox2 constructs we compared the genes affected to those identified as targets of endogenous Sox2 via Chip-Seq analysis. Comparison to the published studies in mouse ESCs and NSCs revealed that 97 of the 165 genes downregulated in our studies were identified as direct Sox2 targets using Chip-Seq analysis. Together with our observation that the levels of exogenous Sox2 appears to be comparable to the level of endogenous Sox2 (Figure S3), this suggests that, although it’s possible that the endogenous Sox2 could affect some non-physiological targets, SUMOylation of endogenous Sox2 is also likely to play a role in its ability to repress target genes.”

“Since our experiments relied on the function of exogenous Sox2 constructs, we compared the genes affected to those identified as targets of endogenous Sox2 via Chip-Seq analysis. Comparison to the published studies in mouse ESCs and NSCs revealed that 97 of the 165 genes downregulated in our studies were identified as direct Sox2 targets using Chip-Seq analysis. Together with our observation that the levels of exogenous Sox2 appears to be comparable to the level of endogenous Sox2 (Figure S3), this suggests that, although it’s possible that the endogenous Sox2 could affect some non-physiological targets, SUMOylation of endogenous Sox2 is also likely to play a role in its ability to repress target genes” (lines 420-427).

Reviewer 1, major comment 1: “As to activation by SUMOylated versus non-SUMOylated Sox2, the authors suggest, again based on data in Fig.3, that only SUMOylated Sox2 is transcriptionally active (see lines 507-508). The reasoning reported above as to repression data, can be applied equally well to the activation data. Moreover, Fig.1 clearly shows that, when tested on the same reporter gene, carrying Sox2-binding sites, constitutively SUMOylated Sox2 is LESS active than wt Sox2 or non SUMOylatable Sox2.”

As mentioned above, we have now included more details on the difference between the luciferase and RNA seq data.

Reviewer 1 concludes the major comment 1 by stating that “Fig.1 clearly shows that, when tested on the same reporter gene, carrying Sox2-binding sites, constitutively SUMOylated Sox2 is LESS active than wt Sox2 or non SUMOylatable Sox2”. It is important to point out that the data presented in figure 1 only show that SUMOylated Sox2 is less active in activating the reporter gene. As stated, this assay does not give an indication of the repression ability of the constructs. This is consistent with RNA sequencing data, which shows that SUMOylated Sox2 has a major impact on the number of genes that are down-regulated. It also has a lesser impact on genes that are up-regulated, but that do not share a particular known physiological function. 

Reviewer 1, major comment 2: “The genes analysed in Fig.3 represent the complete transcriptome of the tested cells. A proportion of these genes are presumably directly regulated by Sox2; in addition, genes regulated by Sox2 may encode factors which in turn affect the expression of genes non directly controlled by Sox2 (for a knock-out experiment in mouse neural cells reporting expression data, see Bertolini J et al., 2019, Cell Stem Cell, PMID: 30849367). The authors should carefully consider these points in Discussion. Do transfected Sox2 constructs affect, in their experiments, only some of the physiological targets of Sox2? Overexpression experiments have the drawback that genes not regulated, in vivo, by the transcription factor under study, or only moderately affected by it, may be inappropriately activated; this effect is well known in molecular oncology. So, the question is whether in their experiments the transfected constructs encoding Sox2 and its SUMOylatable or non SUMOylatable variants, activate/repress the real Sox2 targets. The authors might consult the data in Bertolini et al (see above) and ChIP-Seq for Sox2 available in the literature; see also D’Aurizio R.et al., 2022Int.J.Mol.Sci., PMID:35887306, who compare mouse with human data and provide many references to papers regarding ChIP-Seq).Defining which ones, among the repressed cell cycle-related genes, are indeed regulated in vivo by Sox2 would be very important.”

We think this is a valid point and we would like to thank reviewer 1 for highlighting this. We have now compared our RNA sequencing data with published ChIP sequencing data and added the results of this analysis to the paper. We are pleased to report that this confirms that a high proportion of the genes we identified as repressed by both exogenous wt Sox2 and Sox_K247R-SUMO2, and in particular cell cycle related genes, have indeed been identified in these ChIP-Seq data. The following sections have been added to the paper:

“By comparing the effects of over-expressing exogenous wt Sox2, non-SUMOylatable Sox2 mutant or constitutively SUMOylated Sox2 in human neural stem cells (hNSC), we found that SUMOylated Sox2, similarly to wt (SUMOylatable) Sox2, caused the downregulation of a set of proliferation related genes. Many of these genes have been published as Sox2 targets in Chromatin Immunoprecipitation Sequencing (ChIP-Seq) assay” (lines 119-123).

“Since our experiments relied on the function of exogenous Sox2 constructs, we compared the genes affected to those identified as targets of endogenous Sox2 via Chip-Seq analysis. Comparison to the published studies in mouse ESCs and NSCs revealed that 97 of the 165 genes downregulated in our studies were identified as direct Sox2 targets using Chip-Seq analysis” (lines 420-424).

“In the same comparison with published Chip-Seq data used above, 22 of the 32 cell cycle-related genes downregulated in our studies were identified as direct Sox2 targets” (lines 462-464).

In addition, Maresca et al. [4] investigate Sox2 activity as a pioneering transcription factor. They show that Sox2 binding keeps the chromatin in an open state which allows other transcription factor to bind to their target genes. This could explain why some of the genes that we identified as differentially regulated are not found in ChIP sequencing data from literature. These genes could be genes that are indirectly regulated by Sox2 as a pioneering transcription factor: they don’t have a Sox2 binding site because they are not directly regulated by Sox2.

In response to reviewer 1 minor comments:

1) “In the cell proliferation experiment of Fig.5, differences are initially detected between day 1 and 2, and slightly increase at day 3; then, curves increase in parallel for 2-3 days, when the curves start approaching a plateau. So, the effects of the transfected constructs seem to be only at the initial stages, then they progressively disappear”. 

As stated in the manuscript, this is due to the cells reaching maximum confluency and therefore not having any space left to grow further. It is also visible in Figure 6b, which shows completely confluent cells at day 7. Moreover, the transfected plasmid and encoded protein are susceptible to degradation over time and therefore it is expected that their effect would gradually decrease over the course of days. To deliver this concept in a clearer manner, we have amended the manuscript as follows: “Statistical comparison between the proliferation rates of the 4 samples showed that the proliferation rate of cells over-expressing either wt Sox2 or Sox2_K247R-SUMO2 was significantly lower than the negative control (pcDNA3) at days 2, 3 and 4 (Figure 6A). These differences tended to disappear around 5-6 days after transfection, presumably due to degradation of the transfected plasmid DNA transfected and its encoded protein and to the cells reaching complete confluency. At day 7 after transfection, there was a slight decrease in the proliferation detected, presumably due to cell death and cell detachment caused by high cell confluency (Figure 6B, panel d7)” (lines 497-503).

2) “See major comment 1. Gene downregulation is observed with transfection of both wtSox2 and rescued mut Sox2 (Sox2_K247R_SUMO2).” 

We are not clear what the reviewer is asking with this comment. Please see our response to major comment 1.

3) “Lines 507-509 I strongly disagree with this comment, See Major Comment 1”. 

We have addressed major comment 1 above. 

4) “Lines 516-520 ‘our hypothesis that Sox2 SUMOylation regulates the protein activity as a transcription factor’: this sentence is unclear. ‘We showed that SUMOylated Sox2 acquires transcriptional repression activity compared to non SUMOylatable mutant Sox2’: not true, see comment 1”. 

We agree that that sentence is unclear and did add little to our discussion, repeating points made elsewhere, so we have removed it. 

5) “Lines 551-552 “Our data suggest that when Sox2 cannot be SUMOylated, it loses its ability to repress a set of cell-cycle regulating target 552 genes, resulting in a higher proliferation rate”. See comments 1 and 2. Also, the higher proliferation rate is very transient. The experiment should better be performed in long-term growing cell cultures”.

Again, this statement repeats what was said elsewhere so it has been removed.

6) “It would be important to have access to Tables listing the top 50 most upregulated and the top 50 most downregulated genes in the transfection experiments reported in Fig.3, together with their expression values, as Transcript per Million, and fold changes”. 

We acknowledge that this is a valid point and we have now added these data to the supplementary data.

Reviewer 2, major point 1: “interference with the SUMOylation pathway, for example by transfection of a dominant negative mutant of the UBC9 enzyme, at least in the reporter assay, would strengthen the conclusions”.

We have performed some preliminary experiments looking and SUMOylation pathway and in particular co-transfecting Ubc9 together with wild-type Sox2 to enhance SUMOylation. However, this did not lead to a detectable increase in endogenously SUMOylated Sox2, probably because SUMOylation is tightly regulated and over-expression of Ubc9 might not be enough to bypass such regulation. For this reason, we designed our experimental approach including constitutively SUMOylated mutant Sox2 constructs. Unfortunately, we are currently not able to perform additional experiments due to closure of the laboratory following Dr. Scotting’s retirement.

Reviewer 2, major point 2: “The authors could complement the localization analysis with a microscopy approach, ideally in living cells expressing fusions of the different mutants to a fluorescent protein; or alternatively, by immunofluorescence of fixed cells (they use an Anti-Myc antibody to detect by Western blot the expressed constructions that have the Myc tag)”.

The suggestion of investigating Sox2/SUMOylated Sox2 localisation through microscopy is indeed appealing. However, we do not currently have possibility undertake this. We believe that our data is of great potential interest for other research groups studying SUMOylation and we agree that it would be very interesting to see if differences in localisation between SUMOylated and non SUMOylated Sox2 can be detected by microscopy.

Reviewer 2, minor point 1: “The authors select transfected cells by FACS for the RNA-seq analysis; however, I could not find any description of selection or identification of transfected cells in the proliferation analysis. How much of the cell population is transfected? They mention that the differences observed ‘tended to disappear around 5-6 days after transfection, presumably due to degradation of the plasmid DNA transfected and its encoded protein.’ In my opinion, identification of transfected cells (for example by expressing the Sox2 variants fused to a fluorescent protein) would strengthen the conclusions.”

Reviewer 2 raises an interesting suggestion regarding the proliferation experiment. Unfortunately, this assay does not allow for expression of fluorescent proteins, as that would interfere with the experimental read-out, which is made by reading the emissions of AlamarBlueTM Cell Viability Reagent. We have performed transfections numerous times on this cell lines using the protocol adopted and we have always reached similar efficiency levels. In addition, the proliferation experiment was repeated three times (three biological replicates) and within each replicate, the experiment was performed on three independent wells for each construct at each time point (three technical replicates within each biological replicate). Therefore, this would compensate for any small change in transfection efficiency between experiments. 

As reviewer 2 suggests, we are adding a paragraph with the most relevant findings at the end of the introduction section. 

-Reviewer 2: “Tables 1 and 2 are the same.”

We have now amended these tables.

Reviewer 2, minor point 5: “In my opinion, data from figures 5 and 6 should be included in one figure, it is not necessary to show the three replicates in the way that are presented in the main figure 6A, and scale bars should be included in Figure 6B”. 

We have amended the picture including only averages of the three biological replicates in Figure 6 and added scale bars to Figure 6A.

Reviewer 3, major point 1: “In Fig. 1, authors showed that K247A and K247R mutants were as active as SOX2 in transcriptional activation of a reporter gene. The Western blot data for expression of various SOX2 mutants should be showed in this figure to interpret the luciferase reporter assay”.

In response to reviewer 3, the Luciferase assay presented in figure 1 was repeated 3 times (3 biological replicates), with 4 technical replicas for each biological replicate, therefore we believe that the results are solid. We are including a western blot (shown below) showing expression of the transfected Sox2 mutant constructs in the supplementary data (Figure S1). This western blot was performed as part of a separate experiment, but we believe is a good indication of successful expression of the Sox2 constructs transfected. Reviewer 3, major point 2: “Since K247A and K247R were highly active in assay in Fig. 1, why it was essentially inactive in transcriptional regulation (only repressed 19 and activated 11 genes) in neural stem cells?”

As stated above in response to reviewer’s 1 major comment, the luciferase activation assay is designed based on a single artificial promoter containing three Sox2 binding sites and can only measure direct activation of this single, artificial promoter in HeLa cells. While this assay provides a good indication to investigate if there are differences in the transcriptional activity of Sox2 mutant constructs compared to the wt protein, it does not show how the protein performs in regulating target genes physiologically. As mentioned above in response to reviewer 1, we have now added the following statement to the paper to make this clearer: “the luciferase reporter assay showed that SUMOylation appears to inhibit the activation activity of Sox2. However, this assay only assessed the ability of Sox2 to activate expression when driven by one particular artificial promoter in HeLa cells and did not assess transcriptional repression. Therefore, the effects of SUMOylation on Sox2 transcriptional activity were further analysed using a genome-wide approach in hNSCs” (lines 350-355).

Reviewer 3, major point 3: “A major concern for the study is that RNA-seq was performed for each type of SOX2 proteins for only one dose. Much of the differences in regulation of gene expression could be attributed to different levels of SOX2 expression. For instance, K247R mutant could be expressed very low in this experiment, and thus had little effect on gene expression. It would be more convincing if two or three different doses of SOX2 were tested for each and data were compared among the samples that have similar level of SOX2 proteins”.

We agree that this could be a concern, however expression of the transfected proteins was assessed through western blot (supplementary figure S1). Unfortunately, repeating this experiment multiple times with different doses of DNA transfected was not doable due to the cost of RNA sequencing. Importantly, we want to point out that the RNA sequencing experiment was repeated three times for each Sox2 construct (three independent biological replicates), therefore generating a total of 12 samples (three vector only, three wild-type Sox2, three Sox2_K472R and three Sox2_K247R-SUMO2) and that the differentially expressed genes shown in figure 3 were present in all three replicates. We apologise if this was not clear in the manuscript, we have added the following statement in the results section: “The experiment was repeated three times (three biological replicates) for each condition (transfection of empty vector, wt Sox2, Sox2_K247R, Sox2_K247R-SUMO2) generating 12 samples, which were analysed by RNA sequencing” (lines 378-380).

Reviewer 3, major point 4: “Authors need to address if SUMOylation influences SOX2 transcriptional activity primarily by affecting SOX2 protein stability or promoting its repression activity or both”. 

Reviewer 3 raises an important point in comment 4. Western blot analysis (figure S1) suggests that there is not a major difference in the expression levels of exogenous wt Sox2 compared to Sox2_K247R and therefore the dramatic difference in transcriptional activity seen both via Luciferase assay and via RNA sequencing is likely due to a difference in activity between these two constructs. The only difference between these two constructs is the point mutation (K247R) that abolishes SUMOylation and therefore we can conclude that the differential transcriptional activity is likely due to SUMOylation. By close inspection of figure S1, it might appear that Sox2_K247R-SUMO2 is expressed at slightly lower levels compared to wild-type Sox2 and Sox2_K247R. This is not clearly evident, especially because of the difference in size between the constructs, which run differently on an SDS-PAGE. However, if this was true, it would be in line with our hypothesis, as despite being expressed at lower level, this construct has a dramatic effect on the regulation of target genes, indicating that indeed SUMOylation plays a pivotal role in regulating Sox2 transcriptional activity. On the other hand, our subcellular localisation data show that Sox2-SUMO2 fusion proteins (Sox2-SUMO2 and Sox2_K247R-SUMO2) are present in the cells at slightly higher levels compared to wild-type Sox2 and Sox2_K247R. This could potentially suggest that SUMOylation stabilises Sox2. However, this difference is not statistically significant. Therefore, we cannot determine whether SUMOylation has an effect on Sox2 protein stability based on the data presented in our study. Indeed, this is a very interesting question that would require more sensitive assays to find a definitive answer.

Reviewer 3, major point 5: “Introduction is too long and some of information could be deleted or moved to discussion.”

We have addressed this by editing the original introduction to make it over 20% shorter and more focussed. However, as suggested by this referee we have then added a section with the most relevant findings at the end, bringing the final length to 85% of the original.

Please note, original uncropped and unadjusted images underlying all blot results are included in Supporting Information.

In conclusion, we believe that the identification of Sox2 target genes involved in the regulation of neural stem cells proliferation is of great interest. Moreover, by showing the involvement of SUMOylation in the regulation Sox2 transcriptional activity, we uncover a novel mechanism that has not been previously investigated in this context.

References

1. Liu YR, Laghari ZA, Novoa CA, Hughes J, Webster JR, Goodwin PE, et al. Sox2 acts as a transcriptional repressor in neural stem cells. BMC Neurosci. 2014;15:95. doi: 10.1186/1471-2202-15-95. PubMed PMID: 25103589; PubMed Central PMCID: PMC4148960.

2. Yesudhas D, Anwar MA, Panneerselvam S, Kim HK, Choi S. Evaluation of Sox2 binding affinities for distinct DNA patterns using steered molecular dynamics simulation. FEBS Open Bio. 2017;7(11):1750-67. Epub 20171009. doi: 10.1002/2211-5463.12316. PubMed PMID: 29123983; PubMed Central PMCID: PMCPMC5666385.

3. Holmes ZE, Hamilton DJ, Hwang T, Parsonnet NV, Rinn JL, Wuttke DS, et al. The Sox2 transcription factor binds RNA. Nat Commun. 2020;11(1):1805. Epub 20200414. doi: 10.1038/s41467-020-15571-8. PubMed PMID: 32286318; PubMed Central PMCID: PMCPMC7156710.

4. Maresca M, van den Brand T, Li H, Teunissen H, Davies J, de Wit E. Pioneer activity distinguishes activating from non-activating SOX2 binding sites. EMBO J. 2023:e113150. Epub 20230911. doi: 10.15252/embj.2022113150. PubMed PMID: 37691488.

5. Hay RT. SUMO: a history of modification. Mol Cell. 2005;18(1):1-12. doi: 10.1016/j.molcel.2005.03.012. PubMed PMID: 15808504.

6. Yau TY, Molina O, Courey AJ. SUMOylation in development and neurodegeneration. Development. 2020;147(6). Epub 20200318. doi: 10.1242/dev.175703. PubMed PMID: 32188601; PubMed Central PMCID: PMCPMC7097199.

---

## [Decision Letter · Decision Letter 1]

31 Jan 2024

SUMO-dependent Transcriptional Repression by Sox2 Inhibits the Proliferation of Neural Stem Cells

PONE-D-23-11587R1

Dear Dr. Marelli,

We’re pleased to inform you that your manuscript has been judged scientifically suitable for publication and will be formally accepted for publication once it meets all outstanding technical requirements.

Kind regards,

Anujith Kumar

Academic Editor

PLOS ONE

Additional Editor Comments (optional):

Reviewers' comments:

Reviewer's Responses to Questions

**Comments to the Author**

1. If the authors have adequately addressed your comments raised in a previous round of review and you feel that this manuscript is now acceptable for publication, you may indicate that here to bypass the “Comments to the Author” section, enter your conflict of interest statement in the “Confidential to Editor” section, and submit your "Accept" recommendation.

Reviewer #2: All comments have been addressed

Reviewer #4: (No Response)

2. Is the manuscript technically sound, and do the data support the conclusions?

Reviewer #2: Yes

Reviewer #4: Partly

3. Has the statistical analysis been performed appropriately and rigorously? 

Reviewer #2: Yes

Reviewer #4: Yes

4. Have the authors made all data underlying the findings in their manuscript fully available?

Reviewer #2: Yes

Reviewer #4: Yes

5. Is the manuscript presented in an intelligible fashion and written in standard English?

Reviewer #2: Yes

Reviewer #4: Yes

6. Review Comments to the Author

Reviewer #2: Although the authors could not perform the experiments suggested that would strengthen the conclusions, they followed my other comments and some suggestions of the other reviewers, amending the pointed mistakes and improving the general quality of the manuscript.

Reviewer #4: The manuscript should be reorganized and formatted for better understanding. Figure resolution and order should be improved, then the manuscript might be considered for publication.

7. PLOS authors have the option to publish the peer review history of their article (what does this mean?). If published, this will include your full peer review and any attached files.

Reviewer #2: No

Reviewer #4: No

---

## [Editor Report · Acceptance letter]

22 Feb 2024

PONE-D-23-11587R1 

PLOS ONE

Dear Dr. Marelli, 

I'm pleased to inform you that your manuscript has been deemed suitable for publication in PLOS ONE. Congratulations! Your manuscript is now being handed over to our production team.

Kind regards, 

on behalf of

Dr. Anujith Kumar 

Academic Editor

PLOS ONE